# Stochastic chain termination in bacterial pilus assembly

Christoph Giese [1] ✉, Chasper Puorger[1,3], Oleksandr Ignatov [1,4], Zuzana Bečárová[1], Marco E. Weber[1,5], Martin A. Schärer[1,2], Guido Capitani[2,6] & Rudi Glockshuber [1]

Adhesive type 1 pili from uropathogenic *Escherichia coli* strains are filamentous, supramolecular protein complexes consisting of a short tip fibrillum and a long, helical rod formed by up to several thousand copies of the major pilus subunit FimA. Here, we reconstituted the entire type 1 pilus rod assembly reaction in vitro, using all constituent protein subunits in the presence of the assembly platform FimD, and identified the so-far uncharacterized subunit FimI as an irreversible assembly terminator. We provide a complete, quantitative model of pilus rod assembly kinetics based on the measured rate constants of FimD-catalyzed subunit incorporation. The model reliably predicts the length distribution of assembled pilus rods as a function of the ratio between FimI and the main pilus subunit FimA and is fully consistent with the length distribution of membrane-anchored pili assembled in vivo. The results show that the natural length distribution of adhesive pili formed via the chaperone-usher pathway results from a stochastic chain termination reaction. In addition, we demonstrate that FimI contributes to anchoring the pilus to the outer membrane and report the crystal structures of (i) FimI in complex with the assembly chaperone FimC, (ii) the FimI-FimC complex bound to the N-terminal domain of FimD, and (iii) a ternary complex between FimI, FimA and FimC that provides structural insights on pilus assembly termination and pilus anchoring by FimI.

Many Gram-negative pathogens use adhesive, filamentous protein complexes anchored to their outer membrane, termed pili, to attach to surface glycans of host cells and initiate infection[1–5]. Type 1 pili and the related P pili belong to the best-characterized pilus systems from uropathogenic *Escherichia coli* strains[6]. Type 1 pili bear a single copy of the lectin (adhesin) FimH at their distal end, which recognizes terminal mannosides in high-mannose type N-glycans of the urothelial receptor uroplakin Ia[7,8]. FimH is bound to one or several copies of the minor subunits FimG and FimF[2,9,10], which, together with FimH, form a short,

linear tip fibrillum (Fig. 1a). The tip fibrillum is connected to the pilus rod, a helical and rigid quaternary structure composed of hundreds to several thousand copies of the main structural pilus subunit FimA[11,12]. Type 1 pilus assembly in vivo follows the chaperone-usher pathway[6,13] and is strictly dependent on two protein catalysts: (i) The periplasmic chaperone FimC accelerates pilus subunit folding to a defined tertiary structure up to $10^4$-fold[14–16], while chaperone-bound subunits in archaic and alternative chaperone-usher pilus systems may only be partially folded[17,18]. (ii) The assembly platform ("usher") FimD in the outer

[1]Institute of Molecular Biology and Biophysics, Department of Biology, ETH Zurich, 8093 Zurich, Switzerland. [2]Laboratory of Biomolecular Research, Paul Scherrer Institute, 5232 Villigen, Switzerland. [3]Present address: Institute for Chemistry and Bioanalytics, University of Applied Sciences and Arts Northwestern Switzerland, 4132 Muttenz, Switzerland. [4]Present address: V.I. Grishchenko Clinic of Reproductive Medicine, Blahovishchenska st.25, 61052 Kharkiv, Ukraine. [5]Present address: Laboratory of Physical Chemistry, Department of Chemistry and Applied Biosciences, ETH Zurich, 8093 Zurich, Switzerland. [6]Deceased: Guido Capitani. ✉e-mail: giesec@mol.biol.ethz.ch

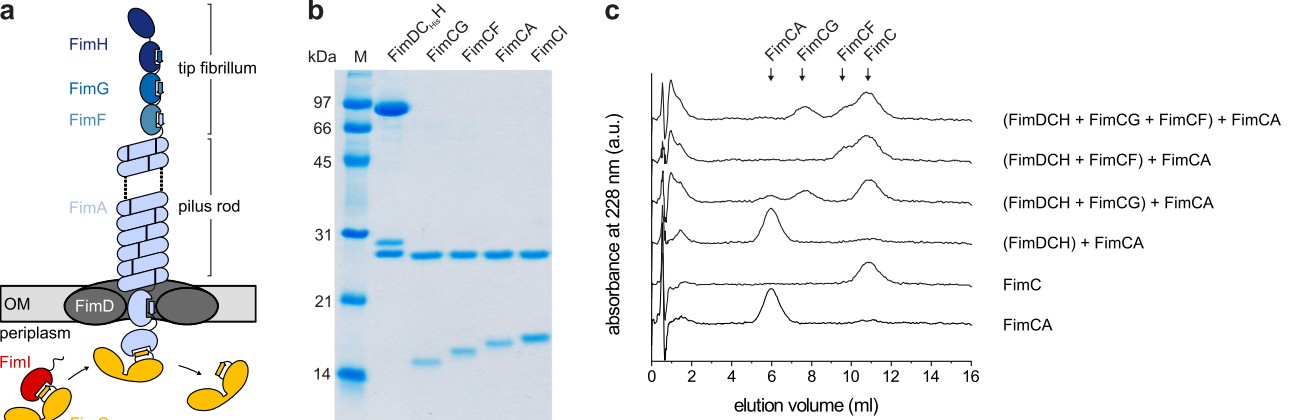

**Fig. 1 | In vitro reconstitution of type 1 pilus rod assembly. a** Architecture and subunit composition of the type 1 pilus. The linear tip fibrillum consists of the adhesin FimH and the minor subunits FimG and FimF. It is attached to the helical pilus rod that may consist of several thousand copies of the main subunit FimA[2,9,11]. Pilus assembly is catalyzed by the assembly platform FimD in the outer membrane (OM)[20]. Only subunits bound to the periplasmic chaperone FimC are assembly-competent and recognized by FimD[27,28]. The mechanism of incorporation of the assembly terminator subunit FimI (red) is the focus of this study. **b** Coomassie-stained polyacrylamide-SDS gel of the purified type 1 pilus subcomplexes (25 pmol each) used in this study. A representative result of three independent repetitions is shown. **c** Analytical cation ion exchange chromatography runs, monitoring the decrease in the concentrations of free FimCA complexes during FimD-catalyzed pilus rod assembly in vitro. FimDCH (0.35 μM) was preincubated with or without an 8-fold molar excess of FimCG or FimCF or both for 5 min at 37 °C. After addition of FimCA (final concentrations of FimDCH and FimCA were 0.25 and 5 μM, respectively) and incubation for 10 min at 37 °C, the reactions were stopped by rapid cooling on ice. The remaining, free FimC-subunit complexes were separated and quantified by cation exchange chromatography at 4 °C and pH 6.7. Reference elution profiles for 5 μM FimC and 5 μM FimCA are shown at the bottom of the panel. Source data are provided as a Source Data file.

membrane catalyzes pilus subunit assembly independently of ATP[19–21]. The pilus subunits undergo a structural rearrangement to a substantially more stable conformation upon assembly, with the released free energy driving polymerization[22–24].

The adhesin FimH is a two-domain protein with an N-terminal lectin domain that binds glycoprotein receptors, and a C-terminal pilin domain that links FimH to the next subunit FimG[7–10,25]. In contrast to FimH, all other pilus subunits are single-domain proteins and structural homologs of the FimH pilin domain, characterized by an incomplete immunoglobulin (Ig)-like fold lacking the C-terminal β-strand[25]. After secretion of the unfolded subunits into the periplasm via the SecYEG system, the oxidoreductase DsbA catalyzes formation of a structural disulfide bond connecting β-strands A and B in each of the type 1 pilus subunits[14]. The periplasmic chaperone FimC then specifically recognizes the disulfide forms of the unfolded subunits and catalyzes subunit folding[14,16]. In all native FimC-subunit complexes, FimC completes the Ig-like subunit fold via donor strand complementation (DSC) by inserting its β-strand G1 into the incomplete Ig fold of the subunit in a parallel orientation relative to the subunit's C-terminal β-strand F[14,25,26]. Except for the distal subunit FimH, all other subunits possess an N-terminal extension (Nte, also termed donor strand) of approx. 15–20 amino acids[25] that remains exposed and unstructured in FimC-subunit complexes.

Only native FimC-subunit complexes are assembly-competent and specifically recognized by the assembly catalyst FimD in the outer membrane[27,28]. FimD consists of five domains: a central, 24-stranded β-barrel transmembrane domain that is obstructed by a plug domain in the inactive, resting state of FimD, an N-terminal and two C-terminal periplasmic domains ($FimD_N$ and CTDs, respectively)[28–35]. Chaperone-subunit complexes are first bound by $FimD_N$[28,36]. Subunit incorporation into the pilus then occurs in an irreversible reaction termed donor strand exchange (DSE), in which the exposed N-terminal donor strand of the incoming subunit, bound to FimC and $FimD_N$, displaces the chaperone capping the last incorporated subunit at the FimD translocation pore[10,12,20,22,25,32,37–40]. In contrast to FimC-subunit complexes, the donor strand of the incoming subunit inserts in the opposite (antiparallel) orientation, which leads to extremely stable subunit-subunit interactions that practically show infinite stability against

dissociation[24]. The donor strand acceptor groove of each subunit possesses five binding pockets, P1–P5, that specifically accommodate hydrophobic side chains of the respective donor strand[12,24,38,39,41]. Structural and biochemical data suggest that the DSE reaction is initiated by threading the donor strand of the incoming subunit into the P5 pocket of the chaperone-bound acceptor subunit[39]. In structures of chaperone-subunit complexes from the type 1, P or Saf pilus system, the P5 position is indeed unoccupied or only transiently occupied by the donor strand of the chaperone[14,25–27,37,39]. In the case of Caf1 pili, however, the donor strand of the chaperone also occupies the P5 pocket[22], indicating that initiation of DSE may not follow the same mechanism across all chaperone-usher pili. During DSE, the incoming chaperone-subunit complex is handed over from $FimD_N$ to the CTDs of FimD, thus resetting $FimD_N$ for binding the next chaperone-subunit complex[29–32,34].

In the case of P pili, it is assumed that assembly termination is achieved when a sixth structural subunit, PapH, is incorporated into the pilus[42,43]. The inhibitory action of PapH is attributed to the observation that the PapH structure, when complexed with the P pilus chaperone PapD, lacks a P5 pocket, which likely prevents donor strand attack and incorporation of another subunit via DSE[43]. Besides its proposed function as assembly terminator, PapH also anchors P pili to the outer bacterial membrane[42]. Despite these insights, termination of pilus assembly by irreversible incorporation of a terminator subunit has never been demonstrated directly for pili assembled via the chaperone-usher pathway, and the mechanism of type 1 pilus assembly termination has remained uncharacterized.

Based on the similar gene arrangements in the P pilus and type 1 pilus gene clusters and the sequence similarity between the *papH* and *fimI* gene, FimI, the fifth structural subunit of type 1 pili, was suggested to act as functional equivalent of PapH[44]. Here, we addressed this hypothesis with an in vitro study, in which we reconstituted the entire type 1 pilus assembly system from all purified components, including the previously uncharacterized subunit FimI. We demonstrate directly that FimI terminates type 1 pilus rod assembly and present a complete, quantitative kinetic description of pilus rod assembly and assembly termination. Specifically, the kinetics of subunit binding to FimD and irreversible subunit incorporation into the growing pilus are in full

agreement with a stochastic chain termination mechanism and reliably predict length distribution histograms of pili formed in vitro and in vivo as a function of the FimI:FimA ratio. Unexpectedly, we find that incorporation of FimI is the fastest assembly step in pilus biogenesis, predicting a high excess of FimA over FimI during pilus assembly in vivo. Moreover, we show that incorporation of FimI is essential for stable anchoring of the pilus to the outer membrane. Finally, we present crystal structures of the FimC-FimI complex, the ternary FimD$_N$-FimC-FimI complex and the ternary FimC-FimI-FimA complex. The latter represents the proximal end of the fully assembled pilus and provides a structural basis for understanding pilus assembly termination and pilus anchoring by FimI.

## Results and discussion

### Initiation of pilus rod assembly requires FimG or FimF at the proximal end of the tip fibrillum

To study the mechanism of the supposed type 1 pilus assembly terminator FimI, we reconstituted the entire FimD-catalyzed type 1 pilus assembly reaction in vitro from all purified components. To this end, we purified the ternary complex between FimD, FimC and FimH (FimDCH) and all binary complexes between the chaperone FimC and the pilus subunits FimG, FimF, FimA and FimI (FimCG, FimCF, FimCA and FimCI; Fig. 1b). First, we probed the ability of FimDCH to catalyze pilus rod assembly from FimCA complexes by mixing FimDCH with a 20-fold excess of FimCA, using either FimDCH alone or FimDCH that had been preincubated with an 8-fold excess of FimCG or FimCF, or a FimCG/FimCF mixture. Kinetics of FimA polymerization were recorded by analytical cation exchange chromatography via the decrease in the concentration of free FimCA complexes and the increase in the concentration of free FimC released upon FimA polymerization (Fig. 1c). While FimDCH alone proved to be inactive as catalyst of FimA assembly, complete turnover of FimCA and release of free FimC was observed when FimDCH had been preincubated with FimCG and/or FimCF prior to the addition of FimCA (Fig. 1c).

Negative-stain electron microscopy confirmed that FimA polymers (pilus rods) had indeed formed when FimDCH had been preincubated with FimCG and/or FimCF (Supplementary Fig. 1). Thus, the initiation of FimD-catalyzed FimA assembly requires the presence of either a FimC-capped FimG or FimC-capped FimF subunit at the growing end of the pilus in the periplasm.

At first glance, the finding that FimDCH alone did not catalyze FimA assembly contradicts a previous study in which FimDCH catalyzed pilus rod assembly from FimCA complexes[20]. Very likely, tiny impurities with FimCG and FimCF in the previous preparation of FimCA are responsible for this discrepancy: Specifically, denatured and dissociated type 1 pili, i.e., a mixture of unfolded FimH, FimG, FimF and FimA, had served as source of FimA for production of FimCA complexes by refolding in the presence of FimC and purification of FimCA with ion exchange chromatography[20]. The resulting FimCA preparation thus may have contained trace amounts of FimCG and FimCF that allowed initiation of FimD-catalyzed pilus rod assembly[20]. In contrast, the FimCA complexes used in the present study were free of contaminations with FimCG and/or FimCF, as they were produced via refolding of FimA from cytoplasmically produced FimA aggregates (inclusion bodies; see "Methods"). In addition, the strict requirement of either FimF or FimG at the growing end of the tip fibrillum for initiation of pilus rod assembly (i.e., the incorporation of the first FimA copy after the tip fibrillum) found here fully agrees with the previous observation that FimD-catalyzed rod assembly was accelerated when purified FimCF and/or FimCG complexes were added[20]. The data also agree with studies on P pilus biogenesis and on type 1 pilus rod assembly in *Salmonella typhimurium*, where assembly of PapA to P pilus rods depended on the incorporation of pilus tip adapter subunits PapF and PapK, and *S. typhimurium* type 1 pilus rod assembly required FimF (the only adapter subunit present in *S. typhimurium*)[45,46].

Together, the dependence of pilus rod assembly initiation on the presence of a minor tip fibrillum subunit appears to be common to all pilus systems with a heterooligomeric tip fibrillum. This mechanism, together with a very slow incorporation of the first FimA subunit (see below), also favors completion of the tip fibrillum prior to rod assembly. In contrast, formation of the chaperone-usher-adhesin complex alone appears to be sufficient for the assembly of pilus rods in pilus systems only bearing a single adhesin at the distal end of the pilus rod, such as for example CS1 or CFA/I pili[47,48]. For pili that lack a dedicated adhesin at their tips, such as Caf1 pili, pilus assembly may be initiated via binding of a ternary chaperone-subunit-subunit complex to the usher[22].

We next analyzed the conditions required for converting FimDCH to the fully active FimA assembly catalyst. For this purpose, we preincubated FimDCH with FimCG, FimCF or both, then added excess FimCA, and measured the kinetics of FimD-catalyzed rod assembly as a function of pre-incubation time (Supplementary Fig. 2). Herein, we refer to this conversion as the "activation of FimDCH", and restrict the term "catalytic activity" to the ability of FimD to catalyze FimA polymerization from FimCA and incorporation of the assembly terminator FimI (see below). When FimDCH was preincubated with FimCG alone, maximum FimD activity was attained after 30 min of preincubation. FimA assembly with 5 µM FimCA and 0.25 µM activated FimDCH took approximately 20 min for completion and FimD activity slowly decreased with longer preincubation times (Supplementary Fig. 2a). With FimCF alone, FimD activity was maximum only after 6 h of incubation, while FimCA assembly was completed in less than 15 min, indicating that FimD was a more efficient catalyst of pilus rod assembly when the first FimA subunit associated with FimF rather than FimG at the growing pilus end. In addition, no loss in FimD activity was observed for even longer preincubation times of up to 20 h (Supplementary Fig. 2b).

After initiation of FimD-catalyzed rod assembly by addition of excess (20-fold) FimCA to the activation reactions, all FimA assembly kinetics exhibited an initial lag-phase, indicating that the rate-limiting step corresponded to the incorporation of the first FimA subunit into the growing pilus[20]. We tested sequential (FimCG first, FimCF second) or simultaneous addition of FimCG and FimCF to FimDCH during the activation reaction and found the latter to yield the highest FimD activity, so that FimA assembly [0.18 µM activated FimDCH, 3.6 µM FimCA (20-fold excess)] was completed about 2-fold faster compared to sequential addition of FimCG and FimCF (Supplementary Fig. 2c). Pili generated with these most active FimD preparations were comparably short with a median length of only $0.6 \cdot 10^2$ nm (Supplementary Fig. 2d). As expected, the increase of the FimCA:FimD ratio from 20:1 to 200:1 reproducibly yielded longer pilus rods with median length of $4.2 \cdot 10^2$ nm (Supplementary Fig. 2e). In the following, we used these conditions for FimDCH activation as standard conditions to analyze the influence of the assembly terminator FimI on the lengths of pilus rods assembled via FimD.

### FimI terminates type 1 pilus assembly

To test whether FimI inhibited pilus rod assembly, purified FimCI (1 µM) was added to the FimD-catalyzed FimA assembly reaction (0.1 µM activated FimDCH, 20 µM FimCA) after 9 min, when rod assembly was completed to about 40% (Fig. 2a). While the assembly reaction in absence of FimCI was completed within 20 min, the reaction stopped within about 5 min after FimCI addition and approximately half of FimCA remained unpolymerized, even after incubation for 65 min (Fig. 2a). The length distributions of the formed pili (200 pili from each reaction) were analyzed by negative-stain electron microscopy. As expected for FimI functioning as the assembly inhibitor and in agreement with about 50% free FimCA complexes left after assembly termination by FimCI, the median pilus length decreased about 2-fold (from $3.8 \cdot 10^2$ nm to $2.1 \cdot 10^2$ nm) when FimCI was added, relative to the reaction in the absence of FimCI (Fig. 2b).

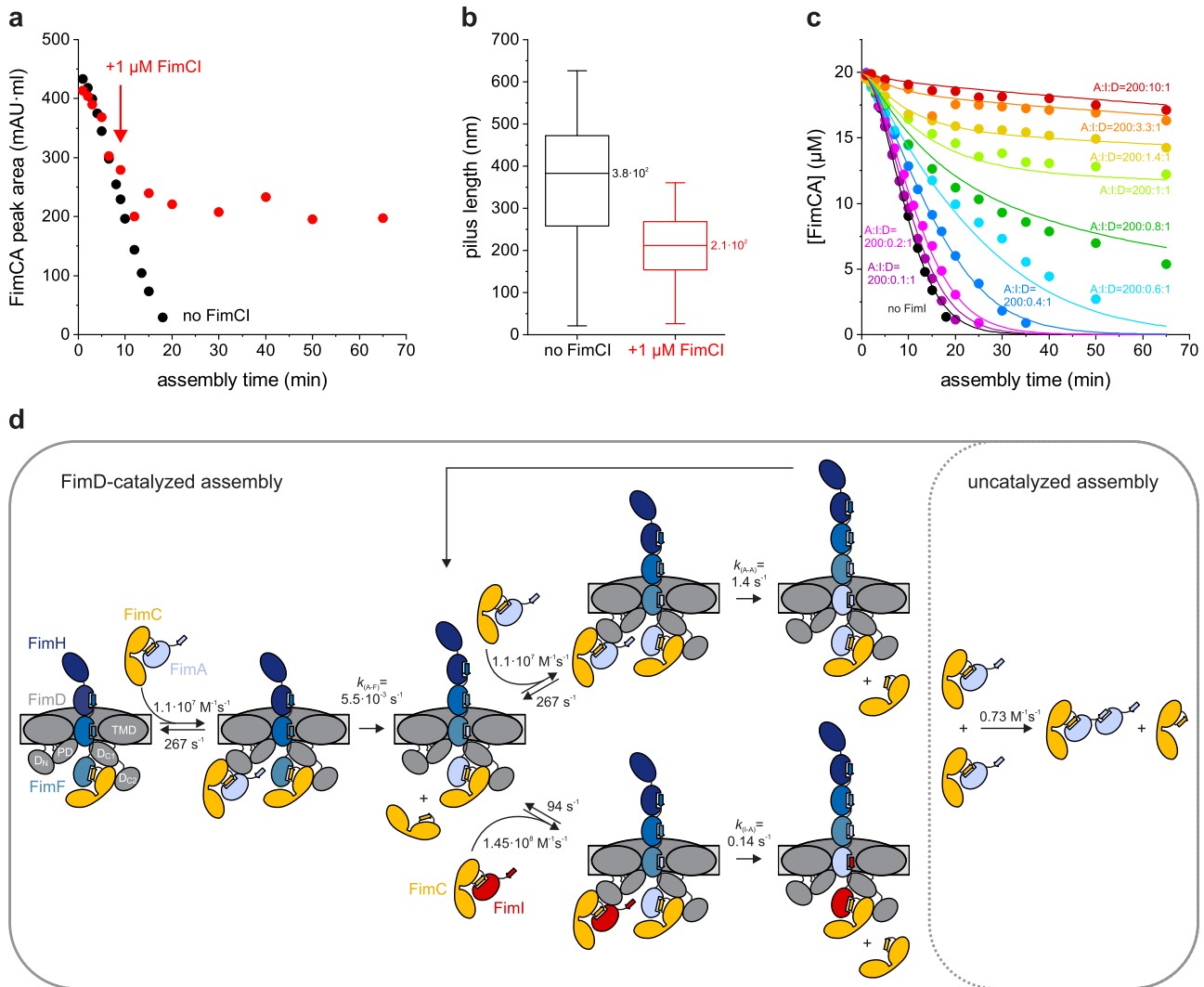

**Fig. 2 | FimI inhibits type 1 pilus rod assembly. a** Kinetics of FimD-catalyzed FimA assembly at pH 8.0 and 23 °C, recorded in absence of FimCI (black) or with FimCI added to 1 μM after nine minutes of the reaction (red). The time point of FimCI addition is indicated by an arrow. FimDCH (0.35 μM) was preincubated with an 8-fold molar excess of FimCG and FimCF for 30 min at 23 °C. After addition of FimCA (final concentrations of FimDCH and FimCA were 0.1 and 20 μM) and further incubation at 23 °C for defined periods of time, the reaction mixtures were analyzed by analytical cation exchange chromatography at 4 °C and pH 6.0. FimA assembly was monitored by recording the decrease in FimCA peak area with time. **b** Box plots of pilus length distributions of the two reactions shown in panel (**a**). Samples taken after 65 min of assembly were analyzed by negative-stain electron microscopy and the length of 200 pili each was measured. The box encloses the second and third quartile, the horizontal line indicates the median and the whiskers the smallest and largest value of the distribution ($n = 200$). **c** Kinetics of FimD-catalyzed FimA assembly in presence of different FimCI concentrations at pH 8.0 and 23 °C. Molar ratios between FimCA, FimCI and FimD (A:I:D) are indicated. Initial concentrations were 0.1 μM FimD, 20 μM FimCA and 0.01–1 μM FimCI. Solid lines show the result of a global fit of the data according to the model depicted in (**d**). **d** Minimal mechanism for type 1 pilus rod assembly and assembly inhibition in vitro. The rate constants for binding/dissociation of FimCA and FimCI to/from FimD$_N$ and for uncatalyzed FimA assembly were kept fixed during fitting. Rate constants for FimCA had been measured previously[51]. The domains of FimD are denoted: N-terminal (D$_N$), plug (PD), transmembrane (TMD) and C-terminal domains (D$_{C1}$, D$_{C2}$). Source data are provided as a Source Data file.

## A comprehensive, quantitative model of pilus rod assembly and stochastic assembly termination by FimI

To obtain detailed and quantitative information on the assembly termination activity of FimCI, FimD-catalyzed FimA assembly kinetics were recorded under conditions where pilus rod assembly was initiated by addition of mixtures of FimCA and FimCI to activated FimDCH. Specifically, the initial FimDCH and FimCA concentrations were kept constant (0.1 μM and 20 μM, respectively) and the initial FimCI concentration was varied between 0.01 and 1 μM (Fig. 2c). Indeed, FimA assembly was slowed down progressively with increasing FimCI concentrations. At the highest FimCI concentration tested (1 μM, 10-fold excess over FimD), FimCA assembly could no longer be detected during 65 min, analogous to incubation of FimCA alone (without FimDCH) (Supplementary Fig. 3a, b). The uncatalyzed FimCA

polymerization proceeded extremely slowly and only yielded small FimA oligomers (but no rods) and was analyzed according to an irreversible second-order reaction as previously described[40], yielding the rate constant for spontaneous, uncatalyzed polymerization of FimCA $k_{(A-A)uncat} = 0.73 \pm 0.02 \, M^{-1} \, s^{-1}$.

The minimal kinetic model of FimD-catalyzed incorporation of an individual FimC-bound subunit into the growing pilus includes three rate constants: Binding ($k_{on}$) and dissociation ($k_{off}$) of the incoming FimC-subunit complex to the N-terminal periplasmic domain of FimD (FimD$_N$) and irreversible subunit incorporation and displacement of the FimC molecule that initially capped the growing pilus end ($k_{DSE}$). This is formally equivalent to a Michaelis-Menten-type model[20,49]. Irreversibility of DSE may additionally be facilitated by helical quaternary structure formation of the assembled pilus rod in the

extracellular space[50]. For global fitting of the kinetics of FimCA consumption at different FimCA/FimCI ratios (Fig. 2c), we took into account that incorporation of the very first FimA subunit, i.e., binding to FimF at the pilus base, is slower than that of any subsequent FimA, where FimA-FimA contacts instead of a FimA-FimF contact are formed[20]. In addition, we included the small contribution of the very slow, FimD-independent (uncatalyzed) formation of FimA oligomers to FimCA consumption during the time window of our in vitro assembly experiment (65 min). Based on our systematic FimD activation experiments (Supplementary Fig. 2) and the previously described rate constants of FimD-catalyzed tip fibrillum assembly[49], we assumed that all activated FimD molecules carried a FimC-capped FimF at the growing end of the tip fibrillum. Regarding FimCI incorporation and assembly termination, we assumed that incorporation of a single FimI completely terminates pilus assembly, and that FimI can only bind to a terminal FimA subunit and not to FimF in the activated FimDHGFC complex (Fig. 2d).

To minimize the number of variable parameters in the global fitting of the kinetics of FimCA consumption (Fig. 2c), we determined the still unknown rate constants of association ($k_{on}$) and dissociation ($k_{off}$) for FimCI binding to FimD$_N$ using stopped-flow fluorescence, as described previously[51] (Supplementary Fig. 3c). Similar to the binding of all other chaperone-subunit complexes to FimD$_N$[51], binding of FimCI to FimD$_N$ proved to be very rapid ($k_{on} = 1.45 \cdot 10^8\,M^{-1}\,s^{-1}$), close to the diffusion-limit of ca. $10^9$–$10^{10}\,M^{-1}\,s^{-1}$ [52,53], and highly dynamic ($k_{off} = 94\,s^{-1}$). Notably, the results showed that FimCI bound faster to and dissociated slower from FimD$_N$ than any other FimC-subunit complex (Table 1).

With the experimentally determined $k_{on}$ and $k_{off}$ values for binding of FimCA and FimCI to FimD$_N$ and the rate constant for uncatalyzed polymerization of FimCA ($k_{(A-A)uncat}$) as fixed parameters, all FimA assembly kinetics (Fig. 2c) were then fitted globally according to the model in Fig. 2d. As FimDCH preparations were shown to contain a small fraction of inactive molecules that are unable to assemble a tip fibrillum[49], the total concentration of catalytically active FimDCH molecules was also included as an open fitting parameter. We obtained a value of $82 \pm 1\%$ for the fraction of active FimDCH molecules, in good agreement with the value of 90% reported earlier[49]. As the global fit (solid lines in Fig. 2c) did not show systematic deviations from the experimental data, we could complete our kinetic model of FimC-catalyzed pilus rod assembly (Fig. 2d) with the DSE rate constants for irreversible binding of FimA to FimF ($k_{(A-F)} = (5.5 \pm 0.5) \cdot 10^{-3}\,s^{-1}$), FimA to FimA ($k_{(A-A)} = 1.4 \pm 0.1\,s^{-1}$) and FimI to FimA ($k_{(I-A)} = 0.14 \pm 0.01\,s^{-1}$) (Table 1). The results confirmed that formation of a FimA-FimA contact is two orders of magnitude faster than binding of the first FimA to FimF[20].

The FimC-subunit complexes FimCG, FimCF, FimCA and FimCI can be considered alternative substrates of FimD. Using the $k_{on}$ and $k_{off}$ values for binding of the different FimC-subunit complexes to FimD$_N$ and the DSE rate constants $k_{(G-H)}$, $k_{(F-G)}$, $k_{(A-F)}$, $k_{(A-A)}$ and $k_{(I-A)}$ determined in this study and previously[20,49], we calculated their apparent $K_M$

values ($K_M^{app} = \frac{k_{off} + k_{DSE}}{k_{on}}$) and specificity constants $k_{DSE}/K_M^{app}$ for the catalyst FimD (Table 1). Notably, the data showed that FimCI is the best substrate of all FimC-subunit complexes, even exhibiting a 4-fold higher specificity constant than FimCA. Overall, the specificity constants obtained for the different FimD substrates ($2.3 \cdot 10^2$ to $2.2 \cdot 10^5\,M^{-1}\,s^{-1}$) are in the range of enzymes with medium catalytic proficiency (Table 1).

Negative-stain electron microscopy of samples taken from the assembly reactions (Fig. 2c) after 65 min of incubation confirmed that, as expected for pilus rod assembly termination by stochastic incorporation of FimI, the length of the assembled pilus rods decreased with increasing FimCI concentrations when FimCA was kept constant (Fig. 3a). Pilus length distributions revealed that the majority of the population gradually shifted to shorter lengths and consequently the median pilus length steadily decreased, from $3.8 \cdot 10^2$ nm for catalyzed assembly in the absence of FimI to $0.4 \cdot 10^2$ nm for assembly in presence of a 3.3-fold excess of FimCI over FimD (Fig. 3b). Figure 3a, b also show that two different pilus rod populations were obtained when substoichiometric concentrations of FimCI relative to FimD were used (in particular at FimCI:FimD ratios of 0.4:1, 0.6:1 and 0.8:1), so that some pilus rods assembled via FimD could not be terminated by FimCI: Besides FimCI-terminated pili shorter than 200 nm, a second population of much longer pili with lengths of up to ~1700 nm was observed under these conditions. For a comparison, the length of pili assembled in the absence of FimCI did not exceed 700 nm (Fig. 3b). We interpret this result such that a larger pool of FimCA complexes was available for pilus rod assembly via the remaining, active FimD molecules when all FimCI complexes had been consumed and had inhibited the other FimD molecules at the early stage of the assembly reaction. In fact, even at the lowest FimCI concentrations used, longer pili already began to appear. The fraction of pili with lengths above 600 nm increased from about 1.5% in absence of FimI to 6.5% and 18% at FimCI:FimD ratios of 0.1:1 and 0.2:1, respectively (Fig. 3b).

Having a complete kinetic model of pilus rod assembly and assembly inhibition at hand (Fig. 2d) allowed us to perform Monte Carlo simulations of FimD-catalyzed pilus rod assembly reactions at different FimCI/FimCA ratios and to generate simulated pilus length distributions for comparison with the experimental data (Fig. 3c). The simulations were performed for the assembly reactions in Fig. 2c, each with 2000 activated FimDHGFC molecules. Overall, pilus rods assembled in silico were found to be slightly shorter compared to the experimental data (median length values were between 57% and 93% of those found experimentally). Nevertheless, the simulated length distributions were qualitatively in full agreement with the experimental distributions in that the median length consistently decreased with increasing FimCI concentration (from 252 nm for assembly in the absence of FimCI to 24 nm for assembly in presence of a 3.3-fold excess of FimCI over FimD). In addition, at substoichiometric concentrations of FimCI relative to FimD, a fraction of up to 19% of the pili exceeded the maximum length of ~400 nm in the simulated length distribution

## Table 1 | Kinetic constants and apparent specificity constants for FimD-catalyzed assembly of FimC-subunit complexes at pH 8.0

| FimD substrate | $k_{on}$ ($M^{-1}\,s^{-1}$)[a] | $k_{off}$ ($s^{-1}$)[a] | $k_{DSE}$ ($s^{-1}$)[b] | $K_M^{app} (= \frac{k_{off} + k_{DSE}}{k_{on}})$ ($\mu M$) | $k_{DSE}/K_M^{app}$ ($M^{-1}\,s^{-1}$) | $\frac{k_{DSE}/K_M^{app}}{k_{DSE}/K_M^{app}(FimCI)}$ |
|---|---|---|---|---|---|---|
| FimCI | $1.45 \cdot 10^8$ | 94 | 0.14 (I-A) | 0.65 | $2.2 \cdot 10^5$ | 1.0 |
| FimCG | $2.85 \cdot 10^7$ | 855 | 2.85 (G-H) | 30.1 | $9.5 \cdot 10^4$ | 0.44 |
| FimCA | $1.1 \cdot 10^7$ | 267 | 1.4 (A-A) | 24.4 | $5.7 \cdot 10^4$ | 0.27 |
| FimCF | $1.8 \cdot 10^7$ | 142 | 0.053 (F-G) | 7.9 | $6.7 \cdot 10^3$ | $3.1 \cdot 10^{-2}$ |
| FimCA | $1.1 \cdot 10^7$ | 267 | 0.0055 (A-F) | 24.3 | $2.3 \cdot 10^2$ | $1.1 \cdot 10^{-3}$ |

[a]$k_{on}$ and $k_{off}$ values refer to binding of FimC-subunit complexes to FimD$_N$. Values for FimCG were taken from[49], values for FimCA and FimCF from[20].
[b]$k_{DSE}$ values refer to formation of preferred subunit-subunit contacts, i.e., incorporation of FimG into FimH (G-H), FimF into FimG (F-G), FimA into FimF (A-F), FimA into FimA (A-A) and FimI into FimA (I-A).

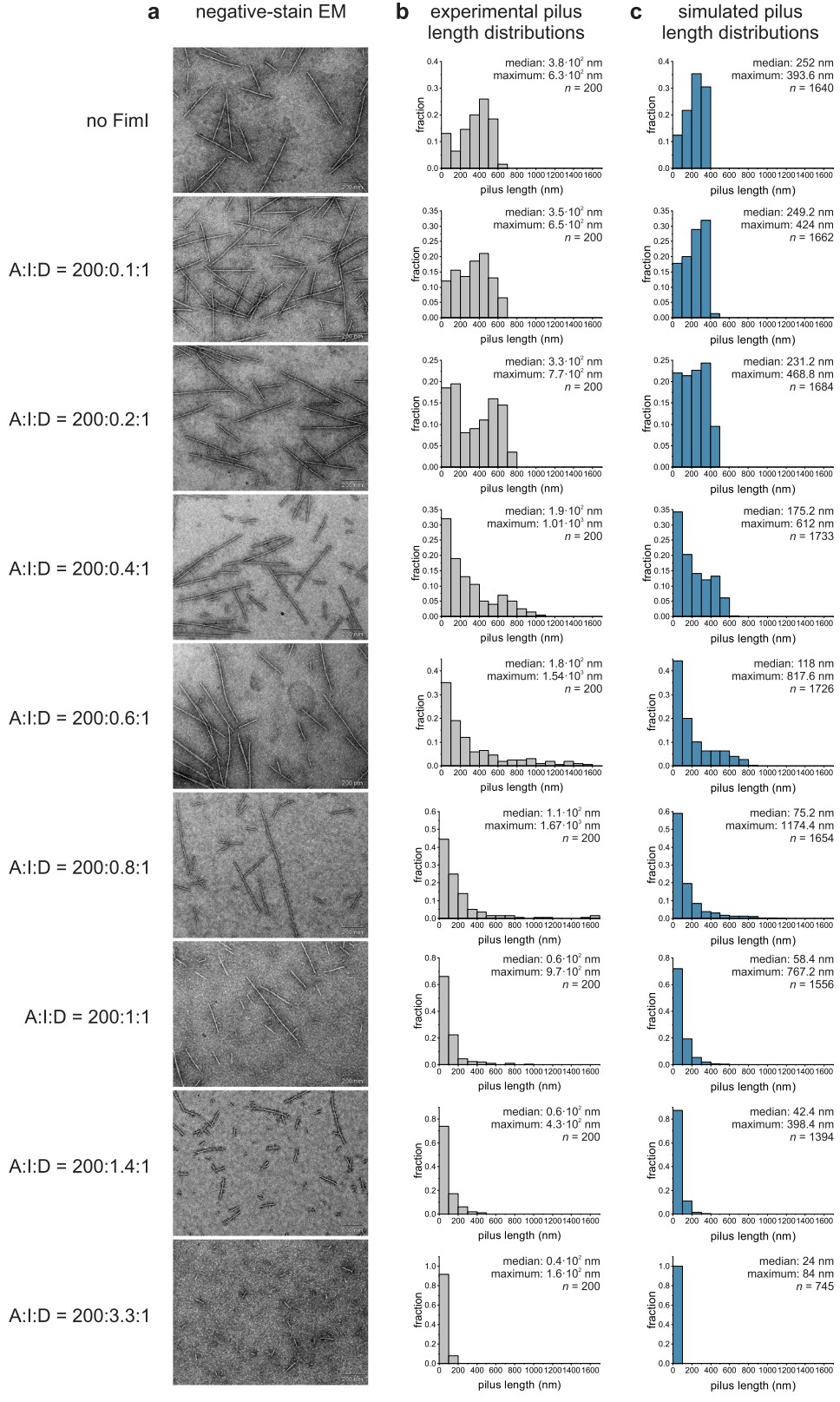

**Fig. 3 | FimI modulates pilus length: Experimental in vitro data versus simulations in silico. a** Representative negative-stain electron micrographs of samples taken from the pilus rod assembly reactions (Fig. 2c) after 65 min of incubation (scale bar = 200 nm). The ratios between FimCA, FimCI and FimD (A:I:D) are indicated on the left and refer to the total initial concentrations of the respective proteins, not taking into account that only 82% of the FimD molecules were catalytically active. **b, c** Length distributions of pili assembled in vitro (gray) and in silico (blue). Median, maximum and size *n* of each sample are indicated. Source data are provided as a Source Data file.

in absence of FimCI (Fig. 3b, c; see also above). Thus, within experimental error, the Monte Carlo predictions are in full agreement with our mechanistic model of pilus rod assembly termination by stochastic incorporation of FimI.

Furthermore, the simulation of pilus rod assembly made it possible to distinguish FimI-terminated pilus rods from non-terminated rods. The simulated length distributions, which included all pili, could therefore be deconvoluted into their two underlying subdistributions of pili with or without FimI bound (Supplementary Fig. 4). As one would expect, at the lowest FimCI:FimD ratio of 0.12:1, the entire distribution was still dominated by pilus rods that had no FimI bound. At higher FimCI:FimD ratios of 0.49:1 and 0.73:1, however, the vast majority of the pili shorter than 200 nm had FimI bound (80% and 95%, respectively), while most pili longer than 200 nm still had no FimI bound (90% and 72%, respectively). This observation agrees with FimCI being the best FimD substrate (Table 1), as this favors formation of predominantly short pili terminated with FimI (note that the fraction of pili shorter than 100 nm is always highest in all the subdistributions of FimI-terminated pili (Supplementary Fig. 4)). The simulations also confirmed that those rods that were (i) assembled at substoichiometric amounts of FimCI relative to FimD and (ii) exceeded the maximum rod length of ~400 nm for assembly simulated in absence of FimI (see topmost panel in Fig. 3c), were predominantly not capped by the assembly terminator FimI. Importantly, these longer pili disappeared at FimCI:FimD ratios above 1:1 in both the experimental data and the simulations (Fig. 3b, c), indicating that all active FimD molecules became inhibited in the presence of equimolar amounts of FimCI. The deconvoluted length distributions further supported this, showing that practically all pili were indeed capped by FimI at the FimCI:FimD ratios 1.2:1, 1.7:1 and 4.0:1 (Supplementary Fig. 4).

In summary, our data show that the natural pilus length distribution results from a stochastic chain termination reaction in which the ratio between FimCA complexes and the FimCI termination complex dictates the pilus length distribution, in addition to the molar FimCA:FimD and FimCI:FimD ratios. The mechanism of PapH, the subunit terminating assembly in the related P pilus system[42,43], is likely identical to that of FimI.

## In vivo modulation of pilus length via the FimCA:FimCI and FimCI:FimD ratios

Type 1 pili attached to the outer *E. coli* membrane show an average length of a few hundred nanometers (see refs. 9,54 and below). Our pilus length distribution analysis revealed that 91% of the pili remained shorter than 200 nm at a 140-fold excess of FimCA over FimCI (Fig. 3b, second-last panel). Thus, despite the fact that transcription of the *fimI* and *fimA* genes is controlled by the same promoter[55] (Supplementary Fig. 5a), for pilus assembly to proceed to the lengths observed in vivo, the concentration of FimCI in the periplasm must be at least two orders of magnitude lower than that of FimCA. Expression of the *fimI* and *fimA* genes is therefore expected to be differentially regulated.

A prerequisite for FimC-catalyzed subunit folding is the DsbA-catalyzed formation of a single, conserved disulfide bond in the structure of all subunits[14] (the disulfide oxidoreductase DsbA is the only known catalyst of disulfide bond formation in the *E. coli* periplasm). Therefore, differences between the kinetics or yields of oxidative folding of FimI and FimA could offer one explanation for different periplasmic concentrations of FimCI and FimCA. We tested this hypothesis and determined rate constants for (i) DsbA-catalyzed oxidation of reduced, unfolded FimI or FimA (FimI$_{red}^U$, FimA$_{red}^U$) and (ii) FimC-catalyzed folding of oxidized, unfolded FimI or FimA (FimI$_{ox}^U$, FimA$_{ox}^U$) using stopped-flow tryptophan fluorescence spectroscopy experiments, essentially as described previously[14] (Supplementary Fig. 5b–g). The kinetics of DsbA-catalyzed FimI oxidation measured under pseudo-first-order conditions (6.7-fold excess of oxidized DsbA

over FimI$_{red}^U$) or using equimolar initial concentrations of oxidized DsbA and FimI$_{red}^U$ were fitted globally according to an irreversible, second-order reaction and yielded a rate constant of oxidation of $1.3 \cdot 10^6\,M^{-1}\,s^{-1}$ (Supplementary Fig. 5b). In comparison, DsbA-catalyzed oxidation of FimA$_{red}^U$ was 40-fold slower ($3.4 \cdot 10^4\,M^{-1}\,s^{-1}$, Supplementary Fig. 5e). FimC-catalyzed folding of FimI$_{ox}^U$ and FimA$_{ox}^U$ occurred with similar rates of $7.3 \cdot 10^4\,M^{-1}\,s^{-1}$ and $4.3 \cdot 10^4\,M^{-1}\,s^{-1}$ (Supplementary Fig. 5c, f). Consequently, the determined rate constants predicted similar in vivo half-lives of ~0.4 and ~1 s for the formation of the native FimCI and FimCA complexes, respectively (Supplementary Fig. 5d, g). In agreement with previous results obtained for FimA, FimG and the pilin domain of FimH (FimH$_P$)[14], FimC-catalyzed folding of FimI was specific to disulfide-intact, unfolded FimI, as no fluorescence change was detected when FimC was mixed with FimI$_{red}^U$ (Supplementary Fig. 5c). Moreover, DsbA- and FimC-catalyzed oxidative folding of both FimA and FimI proceeded to completion in vitro, without yield losses due to unspecific protein aggregation. Together, these results show that the kinetics of oxidative folding of FimI and FimA are very similar and therefore are unlikely to be responsible for differences in periplasmic FimCI and FimCA concentrations. To test whether different *fimI* and *fimA* transcript levels could account for the predicted ~$10^2$-fold excess of FimCA over FimCI in the periplasm, we determined the relative *fimI* and *fimA* mRNA levels in *E. coli* W3110 cells by real-time PCR (Supplementary Fig. 5h). The *fimA* transcript level was only ~8-fold higher than that of *fimI* and ~12-fold higher than that of *fimC*, in good agreement with a previous study that used a DNA microarray to determine transcript levels in *E. coli* MG1655 cells[56]. We conclude that additional mechanisms likely contribute to the regulation of the periplasmic FimCA and FimCI concentrations, for example different efficiency of translation or co-translational translocation into the periplasm and/or proteolytic degradation.

The ≥100-fold higher periplasmic level of FimA relative to FimI implied by our in vitro assembly study is strikingly similar to a previous estimate for the PapA:PapH ratio (of at least 100:1) in P pilus biogenesis[42], supporting the idea that the general principles of pilus rod assembly termination and pilus length modulation are very similar for the type 1 and P pilus system. As a single type 1 pilus, on average, contains approximately 300 FimA molecules (see below and Supplementary Table 1) and taking into account the predicted, more than 100-fold lower level of FimI, we can estimate the upper boundary of the periplasmic FimCI:FimD ratio to be roughly unity. Because DSE, and hence assembly inhibition, is irreversible, there is in fact no need for a much larger intracellular FimCI:FimD ratio to ensure quantitative assembly termination in vivo. It therefore appears that the regulation of the intracellular levels of FimD, FimCA and FimCI and the mechanism of FimD-catalyzed pilus biogenesis co-evolved with the natural type 1 pilus length distribution in *E. coli* (see Fig. 4a).

Next, we addressed the question of whether type 1 pilus length would directly depend on the FimCA:FimCI and FimCA:FimD ratios in vivo. Our above results predicted that the average length of pili displayed on cells would increase with increasing FimCA concentrations when the FimCI levels are kept constant. To test this hypothesis, we used the *fimA* deletion strain W3110Δ*fimA*[12], in which base pairs 4–528 of the *fimA* gene had been removed from the genome of *E. coli* W3110 wild-type. Negative-stain electron microscopy of W3110 and W3110Δ*fimA* cells confirmed that cells of the deletion strain no longer displayed type 1 pilus rods on their surface (Fig. 4b). Successful deletion of *fimA* was also verified by real-time PCR (Supplementary Fig. 5h). To complement W3110Δ*fimA* cells and allow production of variable amounts of periplasmic FimCA, we constructed the plasmid pCG1-AC[12] in which periplasmic coexpression of FimA with FimC is under control of the anhydrotetracycline-inducible *tetA* promoter, allowing fine-tuning of FimCA production by varying the concentration of the inducer anhydrotetracycline (aTc) in the growth medium[57].

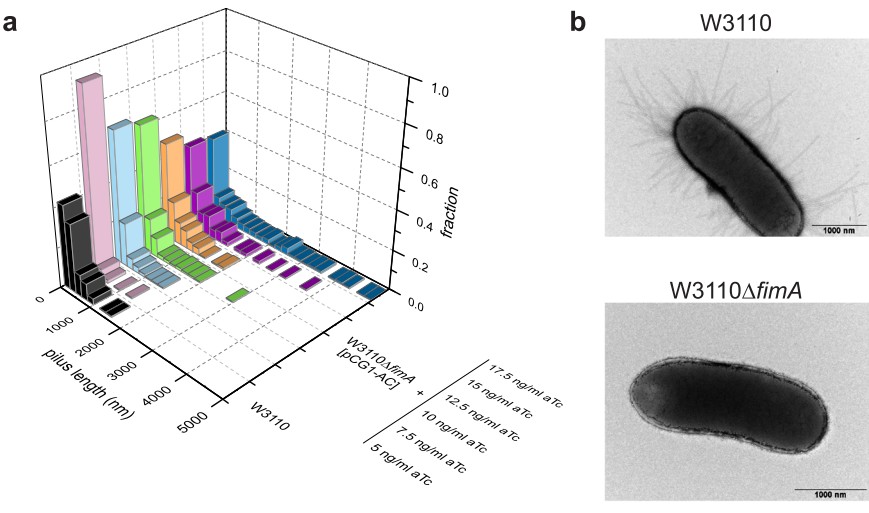

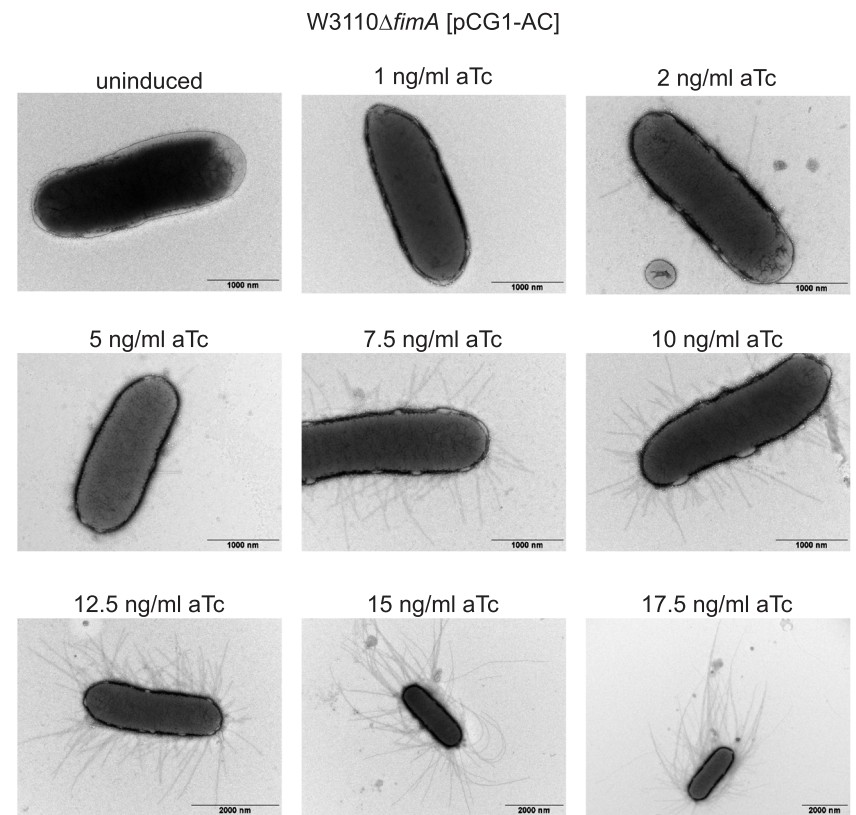

**Fig. 4 | Modulation of type 1 pilus length in vivo. a** Pilus length distributions of W3110 and W3110Δ*fimA* [pCG1-AC] grown in presence of the indicated anhydrotetracycline (aTc) concentrations (*n* = 200). Source data are provided as a Source Data file. **b** Negative-stain electron micrographs of *E. coli* W3110 and W3110Δ*fimA*. **c** Negative-stain electron micrographs of *E. coli* W3110Δ*fimA* harboring plasmid pCG1-AC and grown in absence (uninduced) or presence of the indicated anhydrotetracycline (aTc) concentrations. Representative results of two independent repetitions are shown.

W3110Δ*fimA* cells transformed with pCG1-AC and grown in presence of different aTc concentrations were then analyzed by negative-stain electron microscopy for their pilus length distributions. Figure 4c qualitatively shows that pilus length indeed increased with increasing aTc concentration. For a quantitative analysis, pili were released from cells by a heat step and length distributions were established for 200 pili in each preparation (Fig. 4a and Supplementary Fig. 6). While the median was $2.4 \cdot 10^2$ nm for wild-type cells (corresponding to an average of ~300 copies of FimA per pilus), it increased from $0.7 \cdot 10^2$ nm to $4.0 \cdot 10^2$ nm for W3110Δ*fimA* [pCG1-AC] grown at aTc

concentrations of 5 and 17.5 ng/ml aTc, respectively (Supplementary Table 1). Similarly, the maximum pilus length (defined as the average length of the 5% longest pili) gradually increased from ~0.5 µm (5 ng/ml aTc) to ~4.0 µm (17.5 ng/ml aTc) (Supplementary Table 1). These results show that pilus length in W3110Δ*fimA* [pCG1-AC] can be successfully modulated via the periplasmic FimCA levels (within a relatively narrow range of inducer concentrations). The FimCA:FimCI and FimCA:FimD ratios thus determine pilus length not only in vitro but also in vivo. Our results agree with a previous study on the related P pilus system, where periplasmic overproduction of the main subunit

PapA or the terminating subunit PapH caused an increase or decrease in P pilus length, respectively[42]. Similarly, while disruption of *mrpB*, encoding the terminating subunit MrpB of MR/P pili of *P. mirabilis*, led to significantly longer pili, overproduction of MrpB decreased pilus length in comparison to the parental wild-type strain[58].

### FimI contributes to stable anchoring of type 1 pili to the cell

In the P pilus system, the assembly terminator PapH was also shown to anchor P pili to the outer bacterial membrane[42]. In light of the functional similarity between PapH and FimI suggested earlier[44] and established in this study, we hypothesized that FimI, like PapH, could be involved in anchoring type 1 pili to the bacterial outer membrane. To test this, an *E. coli* W3110 *fimI* deletion strain (W3110Δ*fimI*) was generated. Real-time PCR confirmed successful deletion of *fimI*

(Supplementary Fig. 5h). W3110Δ*fimI* cells were then grown in either static or shaken liquid medium and probed for the presence of functional pili on the bacterial surface by testing the ability of W3110Δ*fimI* to agglutinate with yeast cells bearing highly mannosylated mannoproteins in the cell wall (Fig. 5a). While agglutination readily occurred when W3110Δ*fimI* cells had been grown statically, no agglutination was detected with W3110Δ*fimI* grown in shaken medium, indicating that pili in W3110Δ*fimI* were sheared off mechanically under shaking and lost to the culture medium. A control showed that plasmid-encoded FimI partially complemented *fimI* deficiency of W3110Δ*fimI* and restored membrane anchoring and yeast agglutination.

As expected, wild-type W3110 cells displayed similar piliation levels for both growth conditions and caused agglutination even when grown under shaking. Notably, W3110Δ*fimI* cells grown under shaking

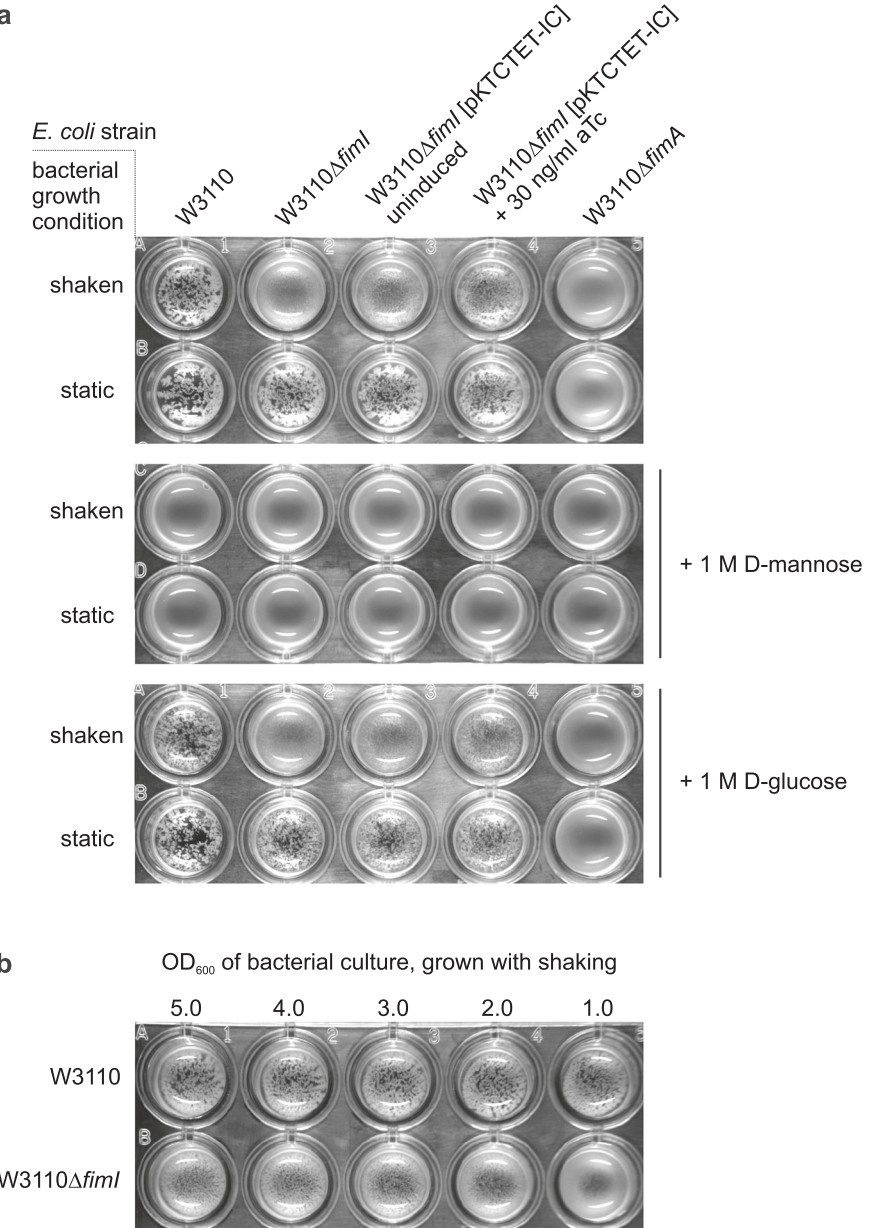

**Fig. 5 | FimI anchors type 1 pili to *E. coli* cells. a** Yeast agglutination assays performed with *E. coli* W3110, W3110Δ*fimI* and W3110Δ*fimA*, which had been grown under shaking or static conditions and diluted to OD$_{600}$ = 1.0 prior to mixing with the yeast cells. To complement the *fimI* deletion, the plasmid pKTCTET-IC was used and gene expression induced with 30 ng/ml anhydrotetracycline (aTc). Mannose specificity of yeast cell binding was tested by a competition assay in the presence of 1 M D-mannose or 1 M D-glucose (negative control). **b** Yeast agglutination assay with *E. coli* W3110 and W3110Δ*fimI*, grown under shaking conditions and diluted to the indicated OD$_{600}$ values prior to mixing with the yeast cells.

also agglutinated weakly with yeast cells when used at higher cell densities in the assay (Fig. 5b). This suggests that not all pili of W3110Δ*fimI* were lost to the culture medium during shaking, but that the loss of pili lowered the agglutination titer of the W3110Δ*fimI* strain compared to that of W3110. A control showed that addition of 1 M D-mannose blocked agglutination throughout, while addition of 1 M D-glucose (the C2 epimer of mannose) did not affect agglutination (Fig. 5a). This confirms that the agglutination assay indeed reported on the presence of functional FimH at the tip of membrane-anchored type 1 pili. W3110Δ*fimA* cells failed to agglutinate yeast irrespective of whether the cells were grown statically or under shaking (Fig. 5a). This agrees with a previous study that demonstrated a strongly impaired adhesion phenotype for *E. coli* cells with inactive *fimA* gene[59]. In the absence of FimA, tip fibrillae may thus still be assembled, but may be too short for reaching enough target mannosides on yeast cell walls. Together, the results show that FimI contributes to anchoring type 1 pili stably to the bacterial outer membrane under shear stress.

## Binding of FimI to the terminal FimA subunit slows dissociation of FimC from FimI 220-fold

The periplasmic, growing ends of type 1 pili are capped with a FimC chaperone that binds to the terminal pilus subunit (FimI in wild-type *E. coli* and FimA in Δ*fimI* strains) via donor strand complementation. The FimC (22.7 kDa) at the growing end is too large to pass the translocation pore of the assembly platform FimD. Release of a pilus to the extracellular medium would require dissociation of periplasmic FimC from the last incorporated pilus subunit. Therefore, FimI at the periplasmic pilus end could contribute to anchoring of the pilus to the outer membrane under mechanical stress by binding particularly strongly to FimC. Likewise, a significant fraction of membrane-associated pili may get lost to the surrounding medium in the absence of FimI. To address this question, we determined the rate constants of spontaneous dissociation of FimC$_{His}$ (FimC with C-terminal hexahistidine tag) from either the binary FimC$_{His}$-FimI (FimC$_{His}$I) or the ternary FimC$_{His}$-FimI-FimAt$_{His}$ complex [FimC$_{His}$IA, (the "t" in FimAt$_{His}$ indicates "N-terminal truncation, without donor strand")] by measuring the time course of competitive displacement of FimC$_{His}$ by excess untagged FimC[40]. While FimC$_{His}$I corresponds to the periplasmic preassembly state of FimI, the ternary FimC$_{His}$IA complex represents the product of the terminal DSE reaction in type 1 pilus assembly.

As a homogeneous FimC$_{His}$IA complex containing FimA wild-type was difficult to prepare due to FimA self-polymerization, we used a truncated FimA variant lacking the N-terminal donor strand (FimAt$_{His}$) to obtain the homogeneous FimC$_{His}$IA complex. FimC$_{His}$I or FimC$_{His}$IA (4 μM) were incubated in the presence of a 9-fold molar excess of untagged FimC (36 μM) at 37 °C and the kinetics of attaining the new equilibrium were monitored with analytical cation exchange chromatography (Fig. 6a, b). Notably, FimC$_{His}$ dissociated from FimC$_{His}$IA 220-fold slower than from FimC$_{His}$I (with dissociation half-lives of 3.85 ± 0.43 h and 63 ± 5.7 s for FimC$_{His}$IA and FimC$_{His}$I, respectively), showing that binding of FimI to the last FimA subunit further stabilizes the FimI·FimC interaction. In addition, association between FimI and FimC$_{His}$ is intrinsically tighter than that between FimA and FimC$_{His}$, because FimC$_{His}$ dissociates about 10-fold faster from FimA (half-life <5 s[40]) than from FimI.

To test whether the slowed FimC$_{His}$ dissociation upon FimI·FimA contact formation in FimC$_{His}$IA was a unique property of FimC$_{His}$IA or is generally observed when a binary chaperone-subunit complex associates with another subunit to a ternary chaperone-subunit-subunit complex, we also measured chaperone dissociation rates for the FimC$_{His}$-FimFt (FimC$_{His}$Ft), FimC$_{His}$-FimF-FimGt (FimC$_{His}$FG), FimCA and FimC-FimA-FimAt (FimCAA) complexes as a control (Fig. 6c, d and Supplementary Fig. 8). While we obtained practically identical FimC$_{His}$ dissociation half-lives for FimC$_{His}$Ft and FimC$_{His}$FG

(1.58 ± 0.14 h and 1.41 ± 0.12 h, respectively), FimC dissociated 40 times slower from FimCAA than from FimCA (dissociation half-lives of 17.7 ± 0.4 min and 26 ± 1 s, respectively). Nevertheless, FimC still dissociated 13 times slower from the FimCIA complex (dissociation half-life of 3.85 h) compared to its dissociation from FimCAA, fully consistent with FimCI's function as pilus assembly inhibitor and anchor in the outer membrane. Notably, for the capsular F1 antigen of *Yersinia pestis*, which consists of only a single type of subunit (Caf1), there is evidence that dissociation of the chaperone Caf1M may also be slower for the ternary Caf1M-Caf1-Caf1 complex than for binary Caf1M-Caf1[23]. Together, these results suggest allosteric stabilization of the proximal chaperone-subunit complex against dissociation through contact formation to the previously incorporated, penultimate subunit.

## Crystal structures of the FimC$_{His}$-FimIt, FimD$_N$-FimC$_{His}$-FimIt and FimC$_{His}$-FimI-FimAt$_{His}$ complexes

As a step towards understanding the structural basis for termination of type 1 pilus assembly by FimI, we solved the crystal structures of the FimC$_{His}$-FimIt (FimC$_{His}$It), FimD$_N$-FimC$_{His}$-FimIt (FimD$_N$CI) and FimC$_{His}$-FimI-FimAt$_{His}$ (FimC$_{His}$IA) complexes to 1.75, 1.7 and 2.8 Å resolution, respectively (Fig. 7a, b, d). In all three structures, the overall folds of FimC and FimI are very similar [Cα root-mean-square deviation (RMSD) values were 1.2 Å (superimposing FimC$_{His}$It with FimCI of FimD$_N$CI), 0.92 Å (superimposing FimC$_{His}$It with FimC$_{His}$I of FimC$_{His}$IA) and 0.95 Å (superimposing FimC$_{His}$I of FimC$_{His}$IA with FimCI of FimD$_N$CI)]. Similarly, the overall structure of FimD$_N$CI closely resembled that of two other ternary FimD$_N$-FimC-subunit complexes, FimD$_N$CF and FimD$_N$CH$_P$[26,27], with Cα RMSD values of 1.4 Å (superimposing FimD$_N$CI and FimD$_N$CH$_P$), 1.4 Å (superimposing FimD$_N$CF and FimD$_N$CH$_P$) and 1.2 Å (superimposing FimD$_N$CI and FimD$_N$CF) (Fig. 7c). Given the high structural similarity of FimC and FimI in the FimC$_{His}$It, FimD$_N$CI and FimC$_{His}$IA complexes, we focus on the analysis of the FimC$_{His}$IA structure in the following.

The asymmetric unit in the FimC$_{His}$IA crystals contained two copies of the FimC$_{His}$IA complex with very similar structures (RMSD of 1.4 Å for 458 aligned Cα atoms). As complex 1 (chains A, B and C) exhibited disorder near the N-terminus of FimAt and in some loop regions, complex 2 (chains D, E and F) was used for structural analysis. The interactions between FimC and FimI in the FimC$_{His}$IA complex were found to be analogous to those observed in the complexes of FimCH, FimCF and FimCA[14,26,27]. Specifically, FimC completes the FimI fold by DSC and inserts residues 101–110 of its G1-strand between the A''- and F-strand of FimI in a parallel orientation with respect to the latter. Leu103, Leu105 and Ile107 of FimC interact with the hydrophobic core of FimI and an additional 22 intermolecular hydrogen bonds, 14 of which are hydrogen bonds between main-chain atoms, are formed between the FimC G1-strand and FimI. Furthermore, FimC residues 1–7 interact with FimI residues 21–26 via six intermolecular hydrogen bonds, two of which are main-chain, and both Arg8 and Lys112 of FimC form a salt bridge with the C-terminus of FimI (Supplementary Fig. 9a).

FimI itself possesses the characteristic, incomplete Ig-like fold with six β-strands, a short α-helical segment (residues 66–69) and a secondary structure topology similar to that of other type 1 or P pilus subunits (Supplementary Fig. 9b, c)[14,24,26,27,43,60]. Consequently, the Ig-like fold of FimI superimposes well onto that of the FimH pilin domain (FimH$_P$), FimF, FimG, FimA and PapH, with pairwise Cα RMSD values between 1.5 and 1.8 Å (Supplementary Fig. 9d). The first resolved residue near the FimI N-terminus in the FimC$_{His}$IA structure was Thr6. Residues 6–20 of FimI constitute the FimI donor strand and insert into the hydrophobic groove between the A- and F-strand of FimAt in an antiparallel orientation compared to FimAt's F-strand. Met13, Phe15 and Ile19 of FimI point towards the center of FimAt and complete its hydrophobic core by occupying binding pockets P2, P3 and P5, respectively (Fig. 7e). Similar to what had been observed in structures

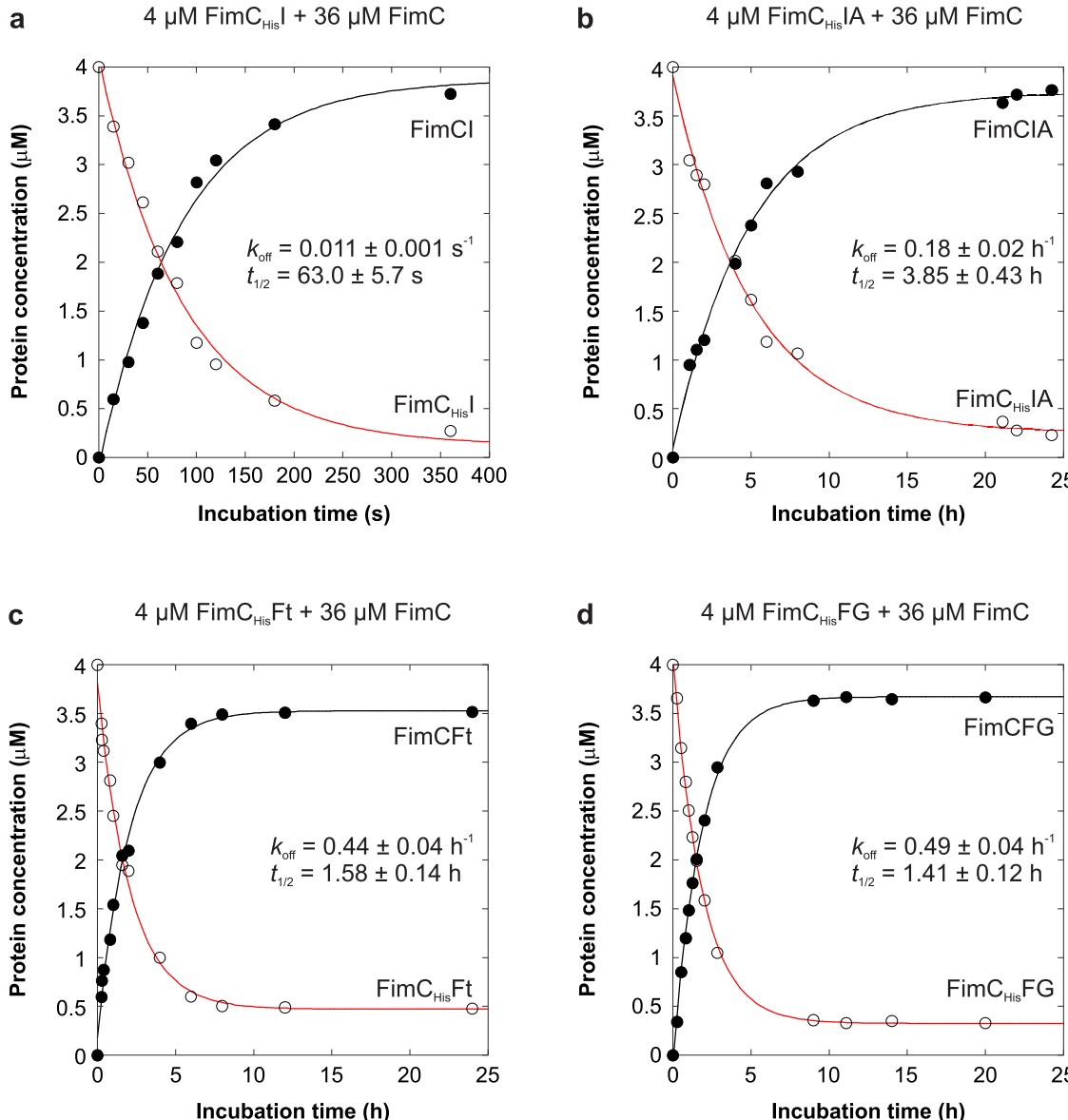

**Fig. 6 | Kinetics of FimC$_{His}$ dissociation from different chaperone-subunit complexes at 37 °C.** For determination of the FimC$_{His}$ dissociation rate constant ($k_{off}$), 4 μM of FimC$_{His}$I (**a**), FimC$_{His}$IA (**b**), FimC$_{His}$Ft (**c**) or FimC$_{His}$FG (**d**) were incubated with a 9-fold molar excess of FimC. Relaxation to the new equilibrium was monitored by analytical cation exchange chromatography. Solid lines are fits according to a single-exponential function. Source data are provided as a Source Data file.

of FimA complemented with the FimA donor strand[61,62], the FimA pockets P1 and P4 are shallow and hence occupied by Gly11 and Gly17 of FimI (Fig. 7e). In addition to these interactions, 31 intermolecular hydrogen bonds are formed between the FimI donor strand and FimAt (24 of which are between main-chain atoms), and Glu22 of FimI interacts with Thr39 and Ala40 of FimAt through another four hydrogen bonds.

Besides these FimC-FimI and FimI-FimAt contacts, the FimC$_{His}$IA complex harbors a third interface. It is located between FimC and FimAt, has an interface area of 351 Å² and involves both domains of FimC that form contacts to the loops connecting β-strands A and B, C and D, and E and F of FimAt (Supplementary Fig. 9e). In this interface, FimC and FimAt interact via a salt bridge between Glu62 of FimC and Arg38 of FimAt, a single hydrogen bond between the side chain of Asn191 of FimC and the main-chain carbonyl oxygen of Ala93 of FimAt, amide-π stacking between Tyr193 of FimC and the Gly146-Ala147 peptide bond of FimA, and through hydrophobic interactions between Ala195 of FimC and Pro145 of FimAt (Supplementary Fig. 9e). These

additional FimC-FimAt interactions may be the reason for the 220-fold slower dissociation of FimC$_{His}$ from FimC$_{His}$IA compared to FimC$_{His}$I, and may prevent the dissociation of FimC from FimI at the pilus base and hence anchor the entire pilus to the membrane.

Previously, X-ray and electron cryomicroscopy (cryo-EM) structures of a pilus assembly intermediate, the complex between FimD and the type 1 pilus tip fibrillum (FimDHGFC), were solved[31,32]. In these structures, FimH is already translocated to the extracellular side of the outer membrane, FimG is located inside the FimD pore, and FimF and FimC are bound to the CTDs on the periplasmic side of FimD. On this structural basis and together with the solved FimC$_{His}$IA structure, we tried to model the FimD-bound structure of the proximal end of the type 1 pilus rod terminated by FimCI. Assuming that FimCI, after donor strand exchange, is transferred from FimD$_N$ to the CTDs of FimD, we superimposed FimC$_{His}$ of the FimC$_{His}$IA complex onto FimC of the FimDHGFC complexes. While the structures of FimI and FimF superimposed rather well, FimAt of the FimC$_{His}$IA complex did not line up with FimG, but clearly clashed with FimD's transmembrane domain

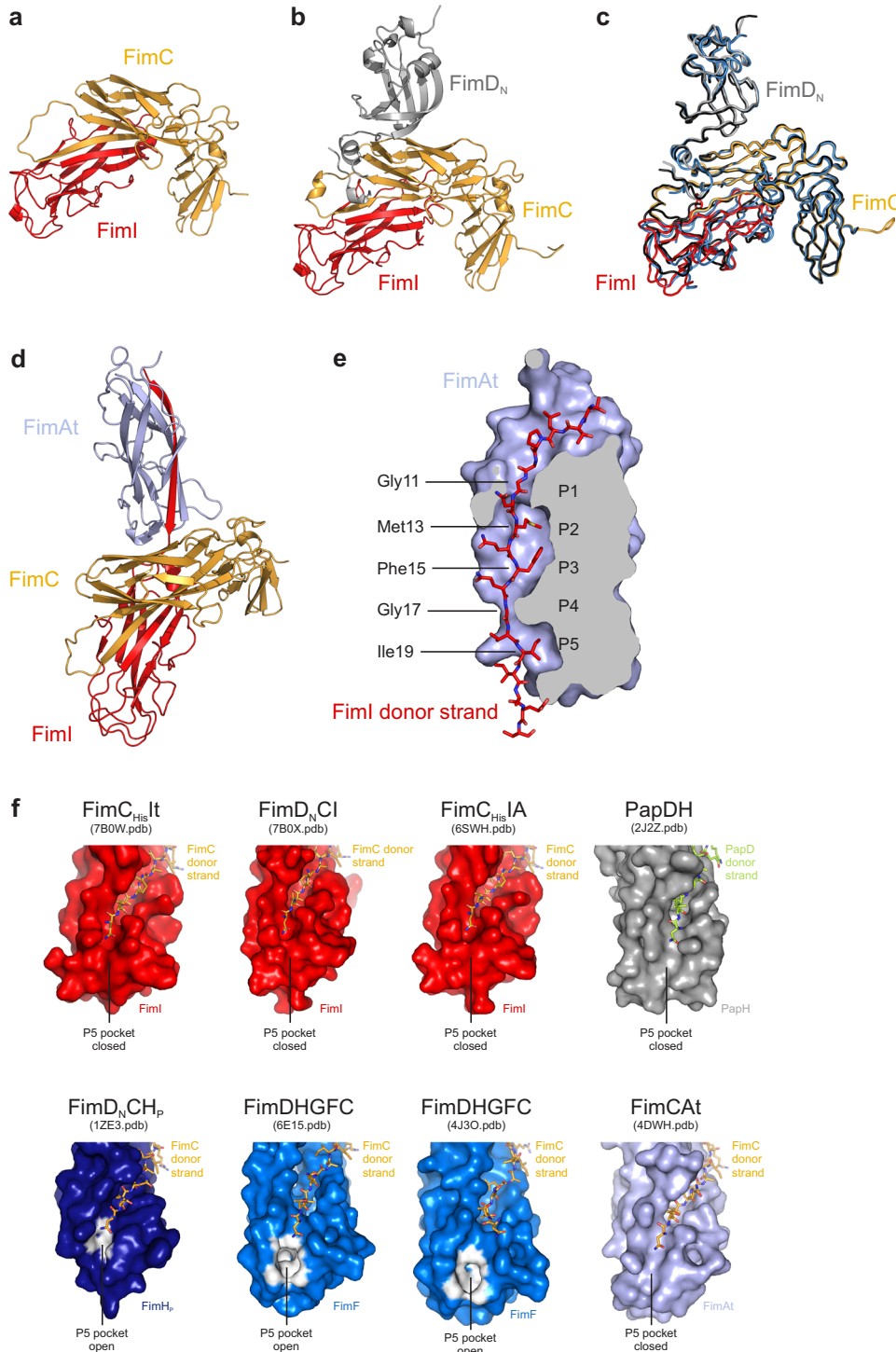

**Fig. 7 | Crystal structures of the FimC_His_It, FimD_N_CI and FimC_His_IA complexes.**
**a** Crystal structure of the binary FimC_His_It complex. Proteins are shown in cartoon representation with FimC in gold and FimI in red. **b** Crystal structure of the ternary FimD_N_CI complex. Proteins are shown in cartoon representation with FimD_N in gray, FimC in gold and FimI in red. **c** Superimposition of the structures of the FimD_N_CI, FimD_N_CF and FimD_N_CH_P complexes. FimD_N_CI is colored as in panel (**b**), FimD_N_CF is in blue and FimD_N_CH_P in black. **d** Crystal structure of the ternary FimC_His_IA complex. All proteins are shown in cartoon representation with FimC in gold, FimI in red and FimAt in light blue. **e** Complementation of the FimAt fold by the FimI donor strand. FimAt is shown in sliced surface representation, the FimI

donor strand is shown as stick model. P1 to P5 indicate the positions in the hydrophobic groove of FimAt where side chains of the FimI donor strand interact with the core of FimAt. **f** Conformation of the P5 pocket in FimI, PapH, FimH_P, FimF or FimAt as observed in the structures of FimC_His_It, FimD_N_CI, FimC_His_IA, PapDH, FimD_N_CH_P, the complex between FimD and the type 1 pilus tip (FimDHGFC), and FimCAt. While analysis of the structures by the CASTp web server[63] did not reveal a distinct P5 pocket in FimI, PapH and FimAt, the P5 pocket is clearly defined in FimH_P of the FimD_N_CH_P complex and in FimF of the FimDHGFC complex. All subunits are shown in surface representation. FimH_P and FimF atoms identified by CASTp to be involved in formation of the P5 pocket are highlighted in white.

(Supplementary Fig. 9f). This discrepancy suggests that either FimD changes its conformation upon incorporation of FimA and formation of the helical quaternary structure of the rod on the extracellular side of FimD (the structure of FimD in complex with an assembled pilus rod is still unknown) or conformational changes occur within FimD and/or FimCIA upon assembly termination by incorporation of FimCI.

For the P pilus system, previous work suggested that assembly termination by PapH is caused by the absence of a P5 pocket in the PapDH structure[43], thus making PapH incapable of accepting donor strands of incoming chaperone-subunit complexes. Analysis of the FimC$_{His}$IA crystal structure by the CASTp web server[63] revealed that FimI, akin to PapH, does not possess a distinct P5 pocket (Fig. 7f). The same analysis revealed that the P5 pocket is in an open conformation in FimH$_P$ of the FimD$_N$CH$_P$ complex (with a volume of 3 Å$^3$), as well as in FimF in the structures of the FimDHGFC complex with volumes of 51.3 Å$^3$ and 18.4 Å$^3$ for the cryo-EM and crystal structure, respectively (Fig. 7f). Therefore, both FimI and PapH may indeed achieve termination of pilus rod assembly by analogous mechanisms that prevent displacement of FimC at the growing pilus end by an incoming donor strand. Notably, however, the P5 pocket of FimA in the structure of the FimCAt complex is also closed (Fig. 7f). Whether or not there is a clear-cut correlation between an open/closed P5 pocket in the FimC-capped subunit at the pilus base and its ability to undergo DSE with another subunit thus still remains to be shown.

In summary, we here identified FimI as the terminating subunit in type 1 pilus rod assembly and presented a quantitative model for rod assembly and its termination that is consistent with the natural pilus length distribution and in full agreement with a stochastic chain termination mechanism. Our results, in particular the in vitro reconstitution of FimD-catalyzed pilus rod assembly and termination, provide a basis for structure determination of FimD bearing an assembled pilus rod on the extracellular side and the FimCI termination complex on the periplasmic side by cryo-EM. The in vitro reconstitution of type 1 pilus assembly also contributes to a better understanding of enzyme-catalyzed assembly of filamentous protein polymers and provides a general framework for testing the mechanism of assembly termination and pilus length regulation in related filamentous pilus systems.

## Methods

### Bacterial strains and plasmids

The *E. coli* W3110Δ*fimI* strain was created by deleting amino acids 43–116 of the mature FimI protein as described[64] using wild-type *E. coli* W3110 cells[65] as parental strain and, unintentionally as a result of the cloning strategy, replacing this amino acid stretch by the sequence CLSLVDG.

The DNA sequence encoding FimI without its signal peptide was amplified by PCR using genomic DNA of *E. coli* W3110 cells as template and cloned into pET-11a[66] yielding the plasmid pFimI_cyt where *fimI* transcription is controlled by the T7*lac* promoter. For periplasmic coexpression of *fimI* and *fimC*, both genes were subcloned from plasmid pACIC-P$_{tet}$[67] into pKTCTET-0[68] via NdeI and SpeI restriction sites. The resulting plasmid was termed pKTCTET-IC and contains a *tetA/T7* tandem promoter, the *tetR, bla, fimI* and *fimC* genes and a pUC origin of replication. Plasmids for periplasmic expression of FimCG and FimCF were obtained by cloning the genes encoding FimG and FimC or FimF and FimC into pTrc99A[69]. Plasmid pfimC_cyt for cytoplasmic expression of FimC was generated by subcloning the *fimC* gene from pfimC$_{His}$_cyt[16] into pET-11a via NdeI and BamHI restriction sites. Plasmid pfimAt_cyt for cytoplasmic expression of FimAt (residues 18–159 of FimA) was generated using the QuikChange II Site-Directed Mutagenesis Kit (Agilent), primers 5′-CTTTAAGAAGGAGATATACATATGA ACGCCGCTTGCGCAGTTG-3′ and 5′-CAACTGCGCAAGCGGCGTTCAT ATGTATATCTCCTTCTTAAAG-3′, and the plasmid encoding FimAt with N-terminal His$_6$-tag[61] as template.

### Protein production

The N-terminal domain of FimD (FimD$_N$, residues 1–125 of FimD), a FimC variant with a C-terminal hexahistidine tag (FimC$_{His}$) and the FimC$_{His}$-FimF (FimC$_{His}$F), FimC$_{His}$-FimFt (FimC$_{His}$Ft) and FimC$_{His}$-FimGt (FimC$_{His}$Gt) complexes were expressed and purified as described[16,27,28,41]. FimAt$_{His}$ (wild-type FimA missing residues 1–13 but with an N-terminal hexahistidine tag) was expressed and purified as described[61].

The ternary complex between FimD, FimC and FimH (FimDCH) was produced as described[29,32]. Briefly, *E. coli* Tuner cells carrying pAN2-Strep and pETS1001 were grown at 37 °C in TB medium containing kanamycin at 30 µg/ml and spectinomycin at 100 µg/ml. At OD$_{600}$ = 1.0, gene expression was induced by adding IPTG to 100 µM and L-arabinose to 0.1% (w/v). Glycerol was added to 0.1% (v/v) and the cells grown for 48 h at 16 °C. Cells were harvested by centrifugation, resuspended in 20 mM Tris–HCl pH 8.0 (30 ml per liter of culture) containing complete EDTA-free protease inhibitor cocktail (Roche) and lysed using a Microfluidizer M-110L (Microfluidics, USA). After centrifugation (10 min, 5000 × g, 4 °C), the supernatant was recovered, sarkosyl added to 0.5% (w/v) and the solution stirred for 5 min at room temperature. Outer membranes were pelleted by ultracentrifugation (1 h, 100000 × g, 4 °C) and resuspended with 9 ml of 20 mM Tris–HCl pH 8.5, 120 mM NaCl supplemented with protease inhibitors (Roche) per gram of membranes using a dounce homogenizer. For solubilization, n-dodecyl-β-D-maltopyranoside (DDM) was added to 1.5% (w/v), the suspension stirred for 30 min at room temperature and insoluble material removed by ultracentrifugation (45 min, 100000 × g, 4 °C). The supernatant was passed over a 5 ml HisTrap HP column (GE Healthcare) equilibrated with 20 mM Tris–HCl pH 8.5, 120 mM NaCl, 0.05% (w/v) DDM (buffer A). The column was washed with buffer A containing 25 mM imidazole and bound protein eluted by a step gradient with buffer A containing 250 mM imidazole. The solution was diluted 2-fold with buffer A, loaded onto an 8 ml Strep-Tactin sepharose column (IBA GmbH) equilibrated with buffer A, washed with buffer A and bound protein eluted with buffer A containing 2.5 mM D-desthiobiotin. The solution was concentrated by ultrafiltration using Amicon Ultra centrifugal filters with 100 kDa molecular weight cutoff (Merck) and passed over a Superdex 200 26/60 gel filtration column (GE Healthcare) equilibrated with 20 mM Tris–HCl (pH 8.0 at room temperature), 50 mM NaCl and 0.05% (w/v) DDM. FimDCH eluted in two main peaks, of which the last one was pooled, concentrated to approx. 1 µM, aliquoted, flash-frozen in liquid nitrogen and stored at −80 °C.

For the production of the FimC-FimF (FimCF) and FimC-FimG (FimCG) complexes, *E. coli* HM125 cells carrying pfimF-C-ATG-trc or pfimGC-trc were grown at 30 °C in 2YT medium containing ampicillin at 100 µg/ml. At OD$_{600}$ = 0.7, expression was induced by adding IPTG to 1 mM and the cells grown further for 4 h. Cells were harvested by centrifugation (10 min, 4200 × g, 4 °C), resuspended in 50 mM Tris–HCl pH 7.5, 150 mM NaCl, 5 mM EDTA, 1 mg/ml polymyxin B sulfate (10 ml buffer per liter of culture) and shaken for 1 h at 4 °C. After centrifugation (30 min, 48000 × g, 4 °C), the supernatant was dialyzed against 20 mM Tris–HCl pH 8.0, centrifuged (15 min, 48000 × g, 4 °C) and loaded onto a 15 ml Q-sepharose FF (GE Healthcare) column equilibrated with the same buffer. For FimCF, the flowthrough was collected and solid ammonium sulfate added to a final concentration of 1.2 M. After centrifugation (10 min, 48000 × g, 4 °C), the solution was loaded onto a 10 ml Phenyl-sepharose HP (GE Healthcare) column equilibrated with 20 mM Tris–HCl pH 8.0, 1.2 M (NH$_4$)$_2$SO$_4$ and bound protein eluted with a linear gradient from 1.2 to 0 M (NH$_4$)$_2$SO$_4$. Fractions containing FimCF were pooled, dialyzed against 20 mM MES-NaOH pH 5.5, loaded onto a Source 30 S (GE Healthcare) column equilibrated with the same buffer and proteins eluted with a linear gradient from 0 to 200 mM NaCl. FimCF typically eluted at a conductivity of approx. 8 mS/cm. The appropriate fractions were pooled,

the solution concentrated by ultrafiltration using Amicon Ultra centrifugal filters with a 10 kDa molecular weight cutoff (MWCO) (Merck) and passed over a Superdex 75 16/60 gel filtration column (GE Healthcare) equilibrated with 20 mM Tris–HCl (pH 8.0 at room temperature), 50 mM NaCl. Fractions containing pure FimCF complex were pooled, aliquoted, flash-frozen in liquid nitrogen and stored at −80 °C. FimCG was eluted from the Q-sepharose FF column with a linear gradient from 0 to 400 mM NaCl. Fractions containing the majority of FimCG were pooled, dialyzed against 20 mM MOPS-NaOH pH 6.7 and loaded onto a 6 ml Resource S column (GE Healthcare) equilibrated with the same buffer. Bound FimCG was eluted with a linear gradient from 0 to 200 mM NaCl, fractions containing FimCG were pooled and solid ammonium sulfate added to a final concentration of 1.2 M. After centrifugation (10 min, $48000 \times g$, 4 °C), the solution was loaded onto a 10 ml Phenyl-sepharose HP (GE Healthcare) column equilibrated with 20 mM MOPS-NaOH pH 6.7, 1.2 M $(NH_4)_2SO_4$ and bound protein eluted with a linear gradient from 1.2 to 0 M $(NH_4)_2SO_4$. Fractions containing FimCG were pooled, the solution concentrated by ultrafiltration using Amicon Ultra centrifugal filters with 10 kDa molecular weight cutoff (Merck) and passed over a Superdex 75 26/60 gel filtration column (GE Healthcare) equilibrated with 20 mM Tris–HCl (pH 8.0 at room temperature), 50 mM NaCl. Fractions containing pure FimCG complex were pooled, aliquoted, flash-frozen in liquid nitrogen and stored at −80 °C. Final yields were 0.5 and 1.7 mg per liter of culture for FimCF and FimCG, respectively. The identity of the proteins was confirmed by ESI-MS: Expected/measured masses for FimC were 22730.1/22729.5 and 22729.0 Da; for FimF: 16166.2/16165.0 Da; and for FimG: 14854.3/14853.0 Da.

FimC was produced by growing *E. coli* BL21(DE3) cells harboring pfimC_cyt at 37 °C in 2YT medium containing ampicillin at 100 μg/ml. At $OD_{600} = 1.0$, expression was induced by adding IPTG to a final concentration of 1 mM and the cells grown further for 4 h. Cells were harvested by centrifugation (10 min, $4200 \times g$, 4 °C), resuspended in 100 mM Tris–HCl pH 8.0, 1 mM EDTA (3 ml buffer per gram of cells) and lysed using a Microfluidizer. After centrifugation (45 min, $48000 \times g$, 4 °C), the supernatant was dialyzed against 10 mM Tris–HCl pH 8.0, centrifuged (15 min, $48000 \times g$, 4 °C) and passed over a Q-sepharose FF (GE Healthcare) column. The flowthrough was collected, dialyzed against 20 mM MES-NaOH pH 6.0, centrifuged as above and loaded onto a Source 30 S (GE Healthcare) column equilibrated with the same buffer. Bound FimC was eluted with a linear gradient from 0 to 200 mM NaCl. Fractions containing FimC were pooled and solid ammonium sulfate added to a final concentration of 1.4 M. The protein solution was applied to a Phenyl-sepharose HP (GE Healthcare) column equilibrated with 20 mM MES-NaOH pH 6.0, 1.4 M $(NH_4)_2SO_4$ and FimC eluted with a linear gradient from 1.4 to 0 M $(NH_4)_2SO_4$. FimC-containing fractions were pooled, dialyzed against water, flash-frozen in liquid nitrogen and stored at −20 °C. The final yield was 40 mg per liter of culture.

The FimC-FimA (FimCA) complex was obtained by first producing FimA separately as inclusion body in the cytoplasm of *E. coli* BL21(DE3) cells as described[61]. Cells were grown in 2YT medium containing ampicillin at 100 μg/ml to $OD_{600} = 1.0$, gene expression was induced by adding IPTG to 1 mM and growth continued for 4 h. Cells were harvested, resuspended and lysed as described above for FimC. Inclusion bodies were prepared and solubilized essentially as described[70]. Briefly, $MgCl_2$ and $CaCl_2$ were added to the lysate to a final concentration of 5 mM each, followed by adding DNase I (Roche, 10 mg/ml freshly dissolved in $H_2O$) to 10 μg/ml and stirring for 30 min at 25 °C. Then, 0.5 volumes of 60 mM EDTA-NaOH pH 7.0, 1.5 M NaCl, 6% (v/v) Triton-X-100 were added. The suspension was stirred for 30 min at 4 °C and centrifuged (30 min, $48000 \times g$, 4 °C). Pelleted inclusion bodies were washed five times with 100 mM Tris–HCl pH 8.0, 20 mM EDTA, solubilized by suspending in 6 M guanidinium chloride (GdmCl), 50 mM Tris–HCl pH 8.0, 1 mM EDTA, 50 mM DTT (20 ml buffer per gram inclusion body), stirred for 1 h at room temperature

and centrifuged (20 min, $100000 \times g$, 20 °C). The supernatant was recovered and passed over a Superdex 200 26/60 column (GE Healthcare) equilibrated with 6 M GdmCl, 20 mM acetic acid-NaOH pH 4.0. Fractions containing FimA were pooled, diluted into 6 M GdmCl, 50 mM Tris–HCl pH 8.0 (final FimA concentration approx. 30 μM) and incubated overnight at room temperature in presence of 0.1 μM $CuCl_2$ to allow formation of the intramolecular disulfide bond of FimA. To quench the oxidation reaction, EDTA was added to a final concentration of 0.5 μM. The protein solution was first concentrated by crossflow filtration using a Vivaflow 200 system (Sartorius) and two Hydrosart cassettes (MWCO: 10 kDa) and further concentrated by ultrafiltration. For refolding and simultaneous complex formation with FimC, denatured FimA was rapidly diluted 30-fold at room temperature into 20 mM $NaH_2PO_4$-NaOH pH 7.0, 200 mM NaCl containing a-2-fold molar excess of FimC and complete EDTA-free protease inhibitor cocktail (Roche). The solution was immediately desalted and the buffer exchanged by passing it over a Sephadex G25 (GE Healthcare) XK 50/20 column equilibrated with 20 mM MES-NaOH pH 5.5. For further purification, FimCA was loaded onto a Source 30 S (GE Healthcare) column equilibrated with the same buffer and eluted with a linear gradient from 0 to 200 mM NaCl. The 1:1 complex eluted at a conductivity of approx. 7 mS/cm, appropriate fractions were pooled, concentrated by ultrafiltration and passed over a Superdex 75 26/60 column (GE Healthcare) equilibrated with 20 mM Tris–HCl (pH 8.0 at room temperature), 50 mM NaCl. Fractions containing pure FimCA complex were pooled, concentrated by ultrafiltration, aliquoted, flash-frozen in liquid nitrogen and stored at −80 °C. The identity of the proteins was confirmed by ESI-MS: Expected/measured mass for FimC was 22730.1/22729.5 Da and for FimA 15825.3/15824.5 Da.

The FimC-FimAt (FimCAt) complex was produced analogously to FimCA. *E. coli* BL21(DE3) cells carrying pfimAt_cyt were used to produce FimAt in form of inclusion bodies, which were isolated, solubilized in a total of 20 ml buffer and purified by size exclusion chromatography in presence of 6 M GdmCl as described above. $Cu^{2+}$-catalyzed formation of the single disulfide bond in FimAt, refolding in presence of FimC and subsequent purification of the FimCAt complex by cation exchange and size exclusion chromatography were done as for FimCA. The FimCAt complex was desalted into 20 mM MOPS-NaOH pH 7.0 (37 °C), concentrated by ultrafiltration, aliquoted, flash-frozen in liquid nitrogen and stored at −80 °C. The identity of the proteins was confirmed by ESI-MS: Expected/measured mass for FimC was 22730.1/22730.0 Da; for FimAt, two species with measured masses of 14156.0 and 14287.0 Da were detected, indicating incomplete cleavage of the initiating, N-terminal methionine (expected mass for the cleaved protein: 14156.5 Da).

Unfolded, disulfide-intact FimA ($FimA_{ox}^U$) was prepared by $Cu^{2+}$-catalyzed air oxidation of the reduced, unfolded protein as described above. The solution was then concentrated by crossflow filtration and passed over a Superdex 200 26/60 column (GE Healthcare) equilibrated in 6 M GdmCl, 20 mM Tris–HCl pH 8.0 to isolate monomeric $FimA_{ox}^U$. To prepare reduced, unfolded FimA, $FimA_{ox}^U$ was incubated in presence of 30 mM DTT for 1 h at 37 °C. DTT was removed by desalting using a Sephadex G25 column (GE Healthcare) equilibrated in 3 M GdmCl, 20 mM Tris–HCl pH 8.0, 0.5 mM EDTA.

The FimC-FimI (FimCI) complex was prepared analogously to FimCA. *E. coli* BL21(DE3) cells carrying pfimI_cyt were used to produce FimI in form of inclusion bodies, which were isolated, solubilized and purified by size exclusion chromatography in presence of 6 M GdmCl as described above. To allow $Cu^{2+}$-catalyzed formation of the intramolecular disulfide bond in FimI by air oxygen, the denatured protein was diluted into 6 M GdmCl, 50 mM Tris–HCl pH 8.0, 0.1 μM $CuCl_2$ to a final concentration of 3 μM and incubated at room temperature overnight. After concentrating the solution by crossflow and ultrafiltration, the FimCI complex was formed by rapidly diluting denatured, disulfide-bonded FimI at 4 °C 30-fold into 20 mM acetic

acid-NaOH pH 5.0 containing a-2-fold molar excess of FimC and complete EDTA-free protease inhibitor cocktail (Roche). FimCI was then dialyzed against 20 mM MOPS-NaOH pH 6.7, centrifuged (10 min, 48000 × $g$, 4 °C), loaded onto a Source 30 S (GE Healthcare) column equilibrated with the same buffer and eluted with a linear gradient from 0 to 300 mM NaCl. Fractions containing FimCI were pooled, concentrated by ultrafiltration and passed over a Superdex 75 16/60 column (GE Healthcare) equilibrated with 20 mM Tris–HCl (pH 8.0 at room temperature), 50 mM NaCl. FimCI-containing fractions were pooled, concentrated by ultrafiltration, flash-frozen in liquid nitrogen and stored in aliquots at −80 °C. The identity of the FimCI complex was confirmed by ESI-MS: Expected/measured mass for FimC was 22730.1/22729.5 Da and for FimI 17169.3/17169.0 Da.

The FimC$_{His}$-FimI (FimC$_{His}$I) complex was prepared similarly. Unfolded, disulfide-intact FimI (300 μM in 6 M GdmCl, 20 mM Tris–HCl pH 8.0, 0.1 mM EDTA) was refolded by rapid, 60-fold dilution in refolding buffer (20 mM MOPS-NaOH pH 6.7) containing a small excess of FimC$_{His}$ over FimI. The refolding mixture was dialyzed at 4 °C against 20 mM MOPS-NaOH pH 6.7 to remove residual GdmCl and applied to a Resource S column (GE Healthcare) to separate FimC$_{His}$I from free FimC$_{His}$. The FimC$_{His}$I complex was eluted by applying a linear gradient from 0 to 0.3 M NaCl.

FimIt, a truncated FimI variant lacking the N-terminal donor strand, was prepared in denatured but disulfide-intact form (FimIt$_{ox}^{U}$) analogously as described above for FimI by using E. coli BL21(DE3) cells carrying pfimIt_cyt. The identity of FimIt was confirmed by ESI-MS: expected/measured mass was 15011.9/15012.5 Da.

Unfolded, disulfide-intact FimI (FimI$_{ox}^{U}$) was prepared by Cu$^{2+}$-catalyzed air oxidation of the reduced, unfolded protein as described above. The solution was then concentrated by crossflow filtration and passed over a Superdex 200 26/60 column (GE Healthcare) equilibrated in 6 M GdmCl, 20 mM Tris–HCl pH 8.0 to isolate monomeric FimI$_{ox}^{U}$. Reduced, unfolded FimI (FimI$_{red}^{U}$) was prepared by incubating FimI$_{ox}^{U}$ in presence of 30 mM DTT for 1 h at 37 °C. DTT was removed by desalting using a Sephadex G25 column (GE Healthcare) equilibrated in 3 M GdmCl, 20 mM Tris–HCl pH 8.0, 0.5 mM EDTA.

The FimC$_{His}$-FimIt (FimC$_{His}$It) complex was prepared by rapid, 60-fold dilution of FimIt$_{ox}^{U}$ into 20 mM MOPS-NaOH pH 6.7 containing a 1.6-fold molar excess of FimC$_{His}$. Final FimIt and FimC$_{His}$ concentrations during refolding were 5 and 8 μM, respectively. Refolding was performed at 4 °C with stirring for 1 h. The protein solution was concentrated by crossflow filtration and dialyzed against 20 mM MOPS-NaOH pH 6.7 at 4 °C overnight. Aggregates were removed by centrifugation (20 min, 48000 × $g$, 4 °C) and the protein solution loaded onto a 6 ml Resource S column (GE Healthcare) equilibrated in 20 mM MOPS-NaOH pH 7.0. Bound FimC$_{His}$It was eluted with a linear gradient from 0 to 300 mM NaCl. Fractions containing FimC$_{His}$It were pooled and dialyzed against 10 mM Tris–HCl pH 8.0.

Oxidized DsbA (DsbA$_{ox}$) was produced by growing E. coli BL21(DE3) pLysS cells transformed with plasmid pDsbA3[71] at 37 °C in 2YT medium containing ampicillin at 100 μg/ml. At OD$_{600}$ = 0.6, gene expression was induced by adding IPTG to a final concentration of 1 mM and the cells grown further for 4 h. Harvested cells were resuspended in 200 mM boric acid-NaOH pH 8.0, 160 mM NaCl, 5 mM EDTA, 1 mg/ml polymyxin B sulfate (10 ml buffer per liter of culture) and stirred for 2 h at 4 °C. After centrifugation (30 min, 48000 × $g$, 4 °C), the supernatant was dialyzed against 10 mM MOPS-NaOH pH 7.0 overnight at 4 °C, and centrifuged again (12 min, 5000xg, 4 °C). The supernatant was loaded onto a 6 ml Resource Q column (GE Healthcare) equilibrated in 10 mM MOPS-NaOH pH 7.0. Bound DsbA was eluted with a linear 0-500 mM NaCl gradient. Fractions containing DsbA were pooled and passed over a Superdex 200 26/60 gel filtration column (GE Healthcare) equilibrated in 20 mM MOPS-NaOH, 150 mM NaCl pH 7.0. DsbA-containing fractions were pooled, dialyzed against 10 mM acetic acid-NaOH pH 4.0 overnight at 4 °C and loaded onto a 6 ml Resource S column (GE

Healthcare) equilibrated in 10 mM acetic acid-NaOH pH 4.0. Bound DsbA was eluted with a linear gradient from 0 to 50 mM NaCl. Fractions containing DsbA were pooled, dialyzed against 20 mM MOPS-NaOH pH 7.0, flash-frozen in aliquots and stored at −20 °C.

## Preparation of ternary complexes

The ternary FimC$_{His}$-FimI-FimAt$_{His}$ (FimC$_{His}$IA) and FimC$_{His}$-FimF-FimGt (FimC$_{His}$FG) complexes were prepared by mixing purified FimC$_{His}$I with FimAt$_{His}$ or FimC$_{His}$F with FimC$_{His}$Gt, respectively. The ratio between the binary chaperone-subunit complexes in both cases was either 1:1, or the complex containing the truncated subunit was used at two-fold molar excess. The total protein concentration in the reactions was kept below 15 μM. After incubation for 24–36 h at 37 °C in 20 mM MOPS-NaOH pH 7.0, the ternary complexes were purified by cation exchange chromatography on a Resource S column (GE Healthcare). The FimC$_{His}$IA complex was eluted with a linear NaCl gradient in 20 mM MOPS-NaOH pH 6.7 and eluted at ~200 mM NaCl. In case of FimC$_{His}$FG, chromatography was performed in 20 mM MES-NaOH pH 5.5, and the ternary complex eluted at ~220 mM NaCl.

To produce the ternary FimC-FimA-FimAt (FimCAA) complex, 2 μM purified FimCA (desalted into 20 mM MOPS-NaOH pH 7.0 (37 °C)) was incubated with 10 μM FimCAt at pH 7.0 and 37 °C overnight. After desalting into 20 mM acetic acid-NaOH pH 4.5, the reaction mixture was loaded onto a 1 ml Resource S column (GE Healthcare) equilibrated with the same buffer and bound protein eluted with a linear, 30 column volume gradient from zero to 200 mM NaCl. Fractions containing the majority of FimCAA were pooled and passed over a Superdex 75 16/60 size exclusion column (GE Healthcare) equilibrated with 20 mM MES-NaOH, 100 mM NaCl pH 6.0 (4 °C). Fractions containing pure FimCAA were pooled, desalted into 20 mM MES-NaOH pH 6.0 (37 °C), concentrated by ultrafiltration, flash-frozen in aliquots and stored at −80 °C. The identity of the proteins was confirmed by ESI-MS: Expected/measured mass for FimC was 22730.1/22730.0 Da; for FimA 15825.3/15824.5 Da; for FimAt, again two species with measured masses of 14155.5 and 14287.0 Da were detected, indicating incomplete cleavage of the initiating, N-terminal methionine (expected mass for the cleaved protein: 14156.5 Da).

The ternary FimD$_N$-FimC$_{His}$-FimIt (FimD$_N$CI) complex was prepared by mixing a 2.6-fold molar excess of purified FimD$_N$ with purified FimC$_{His}$It (final concentrations were 60 and 23 μM, respectively). After stirring the solution for 1 h at 4 °C, proteins were concentrated by ultrafiltration at 4 °C (MWCO: 10 kDa) and passed over a Superdex 75 26/60 gel filtration column (GE Healthcare) equilibrated in 20 mM sodium phosphate pH 7.4, 115 mM NaCl. Fractions containing FimD$_N$CI were pooled and dialyzed against 10 mM Tris–HCl pH 8.0.

## Preparation of fluorophore-labeled FimD$_N$

FimD$_N$ was labeled at its N-terminus with 5/6-carboxyfluorescein succinimidyl ester as described[51]. Excess label was removed and the buffer exchanged to 20 mM Tris–HCl pH 8.0 by ultrafiltration. Labeled FimD$_N$ was then separated from unlabeled FimD$_N$ by anion exchange chromatography and dialyzed against 20 mM Tris–HCl pH 8.0.

## Determination of protein concentrations

Protein concentrations were determined via their absorbance at 280 nm using the following molar extinction coefficients: FimDCH: 202000 M$^{-1}$ cm$^{-1}$; FimCF: 33015 M$^{-1}$ cm$^{-1}$; FimCG: 36000 M$^{-1}$ cm$^{-1}$; FimCA and FimCAt: 26680 M$^{-1}$ cm$^{-1}$; FimCI: 44015 M$^{-1}$ cm$^{-1}$; FimC$_{His}$It: 38450 M$^{-1}$ cm$^{-1}$; FimC$_{His}$Ft: 31230 M$^{-1}$ cm$^{-1}$; FimC$_{His}$Gt: 35070 M$^{-1}$ cm$^{-1}$; FimCAt$_{His}$: 25960 M$^{-1}$ cm$^{-1}$; FimC$_{His}$FG: 44720 M$^{-1}$ cm$^{-1}$; FimC$_{His}$IA: 46930 M$^{-1}$ cm$^{-1}$; FimCAA: 29360 M$^{-1}$ cm$^{-1}$; FimD$_N$CI: 44600 M$^{-1}$ cm$^{-1}$; FimC: 24320 M$^{-1}$ cm$^{-1}$; FimI$_{ox}^{U}$: 21100 M$^{-1}$ cm$^{-1}$; FimI$_{red}^{U}$: 20970 M$^{-1}$ cm$^{-1}$; FimIt$_{ox}^{U}$: 15600 M$^{-1}$ cm$^{-1}$; FimA$_{ox}^{U}$: 3110 M$^{-1}$ cm$^{-1}$; FimA$_{red}^{U}$: 2980 M$^{-1}$ cm$^{-1}$; DsbA$_{ox}$: 23050 M$^{-1}$ cm$^{-1}$.

## Determination of association and dissociation rates

Rate constants for binding/dissociation of FimCI to/from FimD$_N$ were determined at 23 °C in 20 mM Tris–HCl pH 8.0. Fluorescein-labeled FimD$_N$ (final concentration: 0.4 µM) was mixed with different concentrations of FimCI in a SX20 stopped-flow instrument (Applied Photophysics, UK). The reaction was monitored by recording the increase in fluorescence intensity above 515 nm (excitation at 495 nm). The fluorescence traces were globally fitted with Dynafit[72] according to a second-order binding and first-order dissociation reaction.

The dissociation rate constant of FimC$_{His}$ from FimC$_{His}$I, FimC$_{His}$IA, FimC$_{His}$Ft or FimC$_{His}$FG and that of FimC from FimCA or FimCAA was determined as described[40] with minor modifications. FimC$_{His}$I, FimC$_{His}$IA, FimC$_{His}$Ft or FimC$_{His}$FG (initial concentration: 4 µM each) were incubated in presence of a 9-fold molar excess of untagged FimC in 20 mM MES-NaOH pH 6.0 at 37 °C in a shaker (300 rpm). Likewise, 4 µM of FimCA or FimCAA were mixed with 36 µM FimC$_{His}$. After defined reaction times, samples were loaded at 4 °C onto a Resource S column (GE Healthcare) equilibrated in 20 mM MES-NaOH pH 6.0 (for reactions containing FimC$_{His}$I, FimC$_{His}$IA or FimCA), 20 mM MOPS-NaOH pH 6.7 (for reactions containing FimC$_{His}$Ft), 20 mM MES-NaOH pH 5.5 (for reactions with FimC$_{His}$FG) or 20 mM acetic acid-NaOH pH 4.5 (for reactions containing FimCAA), and bound proteins eluted with a linear NaCl gradient. The observed kinetics of chaperone exchange were fitted with the following single-exponential function:

$$c_t = c_\infty + (c_0 - c_\infty) \cdot e^{-\frac{1}{\tau}t} \tag{1}$$

where c$_\infty$ is the final protein concentration at equilibrium, c$_0$ the initial concentration and τ the time constant of the exchange reaction. The rate constant for FimC$_{His}$ dissociation ($k_{off}$) can then be calculated from the observed time constant using the known total concentrations of FimC and FimC$_{His}$ according to:

$$k_{off} = \frac{1}{\tau \left(1 + \frac{[FimC_{His}]_{tot}}{[FimC]_{tot}}\right)} \tag{2}$$

## Kinetics of pilus rod assembly

In vitro reconstitution of pilus assembly was performed at 23 °C (unless otherwise indicated) in 20 mM Tris–HCl pH 8.0, 50 mM NaCl, 0.05% (w/v) DDM. To convert FimD to an active FimA assembly catalyst, 0.35 µM FimDCH complex was first incubated with an 8-fold molar excess of FimCG, FimCF or both. For FimD-catalyzed assembly of FimCA both in presence and absence of FimCI, FimDCH was pre-incubated with both FimCG and FimCF for 30 min as this led to the highest activity of FimD with respect to FimA assembly. After pre-incubation, either FimCA alone, or both FimCA and FimCI were added to initiate pilus rod assembly, leading to final concentrations of 0.1 µM FimDCH, 0.8 µM FimCG, 0.8 µM FimCF, 20 µM FimCA and (if included) 0.01, 0.02, 0.04, 0.06, 0.08, 0.1, 0.14, 0.33 or 1.0 µM FimCI. Reactions were monitored by following the decrease in FimCA concentration over time. At defined assembly times, samples were loaded onto a 1 ml Resource S column (GE Healthcare) equilibrated with 20 mM MES-NaOH pH 6.0 and bound FimCA, FimCG, FimCF, FimC and FimCI eluted with a linear gradient from 0 to 195 mM NaCl over 29.25 ml. The absorbance at 228 nm was detected and corrected for that of a FimCA- and FimCI-free sample (in case of reactions containing both FimCA and FimCI) or for that of a FimCA-free sample (in case of reactions containing only FimCA). The areas of the FimCA and FimC peak were determined using PeakFit and the EMG function (PeakFit 4.12., Systat Software, Inc., San Jose, California, USA) and the FimCA peak area plotted against assembly time. Using Dynafit[72], all FimD-catalyzed kinetics were fitted numerically according the mechanism depicted in Fig. 2d, with the rate constants for

uncatalyzed FimA assembly ($k_{(A-A)uncat}$), binding of FimCA and FimCI to FimD$_N$ ($k_{on}$) and dissociation of FimCA and FimCI from FimD$_N$ ($k_{off}$) fixed to $k_{(A-A)uncat} = 0.73\,M^{-1}\,s^{-1}$, $k_{on}$ (FimCA) $= 1.1 \cdot 10^7\,M^{-1}\,s^{-1}$, $k_{on}$ (FimCI) $= 1.45 \cdot 10^8\,M^{-1}\,s^{-1}$, $k_{off}$ (FimCA) $= 267\,s^{-1}$ and $k_{off}$ (FimCI) $= 94\,s^{-1}$, respectively. Each kinetic trace was first fitted individually with the initial FimCA concentration fixed to 20 µM, the initial concentration of activated FimDCH molecules fixed to 0.1 µM, and the FimCA response value and DSE rate constants $k_{(A-F)}$, $k_{(A-A)}$ and $k_{(I-A)}$ as fitting parameters. For normalization and conversion of FimCA peak areas to FimCA concentrations, experimentally determined FimCA peak areas of each kinetic trace were then divided by the corresponding fitted response value. The normalized data were then fitted globally, now with the FimCA response value fixed to 1, the initial FimCA concentration fixed to 20 µM and the concentration of activated FimDCH molecules and $k_{(A-F)}$, $k_{(A-A)}$ and $k_{(I-A)}$ as fitting parameters.

Uncatalyzed FimCA assembly was measured similarly by (i) omitting FimDCH from the reaction mix and using 1 µM FimCI, the highest FimCI concentration used in the catalyzed assembly reactions or (ii) incubating 20 µM FimCA alone. Both sets of raw data were first individually fitted according to an irreversible, second-order dimerization reaction, normalized using the fit value for the peak area at $t = 0$ and the known initial FimCA concentration, and then fitted globally with shared $k_{(A-A)uncat}$.

Samples for negative-stain electron microscopy were removed after 65 min of assembly, flash-frozen in liquid nitrogen and stored at −20 °C for later analysis.

## Electron microscopy and pilus length measurements

Samples for negative-stain electron microscopy were prepared essentially as described[9]. E. coli strains W3110, W3110ΔfimA and W3110ΔfimA [pCG1-AC] were grown statically at 37 °C in 2YT medium (supplemented with ampicillin at 100 µg/ml in case of W3110ΔfimA [pCG1-AC]). In vivo, type 1 pili are assembled within minutes during the exponential growth phase of the culture but assembly slows considerably when the culture approaches the stationary phase[73]. Therefore, in case of W3110ΔfimA [pCG1-AC], anhydrotetracycline (aTc, Sigma) was added to the growth medium just prior to inoculation to final concentrations of 0, 1, 2, 5, 7.5, 10, 12.5, 15 and 17.5 ng/ml to ensure that FimA and FimC expressed from pCG1-AC would be available for assembly, in particular during the exponential phase of the bacterial growth curve. After 10 h of growth, 2 ml of cell culture were centrifuged (10 min, 2500 × g, 4 °C), the pelleted cells resuspended in 100 µl NaH$_2$PO$_4$-NaOH pH 7.5 and a 3 µl drop adsorbed for 20 s to a carbon-coated copper grid (300 mesh, Quantifoil) that had been glow-discharged for 30 s at a current of 25 mA. Excess liquid was removed by blotting the grid with filter paper, followed by negatively staining the sample for 30 s with a 20 µl drop of 1% (w/v) phosphotungstic acid-NaOH pH 7.6. After removing excess stain with filter paper, the grid was air-dried and images recorded using a Morgagni 268 microscope (FEI) operated at an acceleration voltage of 100 kV and equipped with a 1376 × 1032 pixel CCD camera. Cell-bound pili of W3110 and W3110ΔfimA [pCG1-AC] cells were released from the outer membrane by incubating the remainder of the cell suspension in a water bath at 90 °C for 10 min. After centrifugation (10 min, 2500 × g, room temperature), the supernatant was used for grid preparation as above. For samples produced by in vitro pilus assembly reactions, electron microscopy grids were prepared as above but using 2% (w/v) phosphotungstic acid-NaOH pH 7.2 for staining. To investigate a larger area of the grid and thus avoid bias towards shorter pili, an array of 5 × 5 overlapping micrographs was acquired using the Multiple Image Acquisition function of the Morgagni user interface. These 25 micrographs were then stitched using the Grid/Collection stitching plugin of Fiji[74,75]. In case of W3110ΔfimA [pCG1-AC] grown in presence of 17.5 ng/ml aTc, the investigated grid area was further increased by manually

superimposing four overlapping $5 \times 5$ arrays with CorelDRAW (Corel Corporation). The lengths of 200 pili per sample were measured using TrakEM2[76]. Data were binned, analyzed and plotted using OriginPro 9.1 (OriginLab).

## Kinetics of DsbA-catalyzed oxidation and FimC-catalyzed folding of FimA and FimI

Rate constants for DsbA-catalyzed oxidation of reduced, unfolded FimI and FimA (FimI$_{red}^{U}$, FimA$_{red}^{U}$) were determined at 25 °C and pH 8.0 by 10:1 mixing of oxidized DsbA (5.45 μM in 20 mM Tris–HCl pH 8.0) with FimI$_{red}^{U}$ (8.25 μM in 3 M GdmCl, 20 mM Tris–HCl pH 8.0) or FimA$_{red}^{U}$ (8.25 μM in 3 M GdmCl, 20 mM Tris–HCl pH 8.0) using a SX18.MV stopped-flow mixing instrument (Applied Photophysics, UK). Initial concentrations were 5 μM oxidized DsbA and 0.75 μM FimI$_{red}^{U}$ or FimA$_{red}^{U}$. Reactions were monitored by recording the change in intrinsic fluorescence intensity above 320 nm (excitation at 280 nm). For oxidation of FimI$_{red}^{U}$, fluorescence traces were fitted according to a pseudo-first-order reaction (6.7-fold excess of oxidized DsbA over FimI$_{red}^{U}$) or a second-order reaction with identical initial concentrations (0.75 μM of both oxidized DsbA and FimI$_{red}^{U}$), and then normalized. Normalized data were globally fitted according to an irreversible, second-order reaction and sharing the rate constant for oxidation and the initial FimI$_{red}^{U}$ concentration among the two datasets.

Rate constants for FimC-catalyzed folding of oxidized, unfolded FimI and FimA (FimI$_{ox}$, FimA$_{ox}$) were determined by 10:1 mixing of FimC (5.45 μM in 20 mM Tris–HCl pH 8.0) with FimI$_{ox}^{U}$ or FimA$_{ox}^{U}$ (8.25 μM in 3 M GdmCl, 20 mM Tris–HCl pH 8.0) using stopped-flow fluorescence spectroscopy as above and fitting the recorded traces according to a pseudo-first-order reaction.

## Real-time PCR

*E. coli* strains W3110, W3110Δ*fimA* and W3110Δ*fimI* were grown overnight in LB medium at 37 °C with shaking. RNA was extracted using the RNeasy Mini Kit (Qiagen). Residual DNA was removed by on-column DNase digestion. Total yield of purified RNA was determined by absorbance spectroscopy and extracted RNA diluted to a final concentration of 1 μg/μl. cDNA synthesis was carried out using 1 μg of RNA and the High Capacity cDNA Reverse Transcription Kit (Applied Biosystems) according to manufacturer's instructions. Real-time PCR was performed using TaqMan gene expression master mix and TaqMan gene expression assays (Applied Biosystems) on an ABI 7900 instrument with the following cycling parameters: (1) 50 °C for 2 min; (2) 95 °C for 10 min; (3) 95 °C for 15 s and 60 °C for 1 min, repeated 40 times. Primers used were: for *fimI*: fimI_FW377 (5′-ATGAAGGAAACCT CGTACCG-3′), fimI_RV451 (5′-CGATATTTGGCGATGAAATG-3′) and fimI_probe406 (5′-CCTCCAGCAAACTGGAAACGGC-3′); for *fimA*: fimA_FW134 (5′-CAGTTGATGCAGGCTCTGTT-3′), fimA_RV251 (5′-AGAT GCAACATTGGTATCGC-3′) and fimA_probe 2 (5′-CCTTCCTGTG CCAGCGATGC-3′); for *fimC*: fimC_FW342 (5′-CCGGGAAAGTTTA TTCTGGA-3′), fimC_RV450 (5′-CTAATTTAGCCGGGCGATAG-3′) and fimC_probe (5′-CAGCTCGCAATTATCAGCCGCA-3′); for GAPDH: GAP DH_FW (5′-AGCTGCAACTTACGAGCAGA-3′), GAPDH_RV (5′-CTTTAG CATCGAACACGGAA-3′) and GAPDH_probe (5′-TTCGGTGTAGCCCA-GAACGCC-3′). All probe primers were labeled with FAM and TAMRA at their 5′- and 3′-end, respectively. PCRs were set up in triplicate and repeated twice (corresponding to three biological replicates with three technical replicates each). Data were analyzed with the SDS 2.3 software. Raw $C_T$ values were transformed to relative transcript levels by the $2^{-\Delta C_T}$ method using GAPDH as internal reference gene[77].

## Yeast agglutination assay

Yeast agglutination assays were performed as described[78] with minor modifications. *E. coli* strains W3110, W3110Δ*fimA*, W3110Δ*fimI* and W3110Δ*fimI* [pKTCTET-IC] were grown at 37 °C in LB medium (supplemented with ampicillin at 100 μg/ml in case of W3110Δ*fimI*

[pKTCTET-IC]) under static conditions or with shaking at 210 rpm in a Multitron shaker (Infors HT). After 14.5 h of growth, cells were harvested by centrifugation (10 min $4000 \times g$, 4 °C), resuspended in 20 mM NaH$_2$PO$_4$-NaOH pH 7.0, 150 mM NaCl and diluted to OD$_{600}$ = 1.0 with the same buffer. In 24-well Linbro plates, 250 μl of the bacterial suspension were then mixed with 50 μl of a 10% (w/v) suspension of dry baker's yeast prepared in 20 mM NaH$_2$PO$_4$-NaOH pH 7.0, 150 mM NaCl and imaged after 5 min of incubation. To test for mannose specificity, bacteria were shortly preincubated with 1.2 M D-mannose or D-glucose prior to mixing with the yeast cells.

## Monte Carlo simulations

Monte Carlo simulations of pilus rod assembly reactions in absence and presence of FimCI were performed using MATLAB and its accessory application SimBiology (The MathWorks, Inc., Natick, Massachusetts, USA). To be able to assess the number of FimA monomers incorporated by a given FimD molecule at the end of the simulation, the mechanism depicted in Fig. 2d was modified such that a separate pool for each polymeric and intermediate species was defined (see Supplementary Fig. 7 for a schematic of the first three steps of the polymerization reaction). The maximum possible number of FimA monomers per pilus rod was set to 2200 to ensure inclusion of potentially very long pili. Simulations were run with initial numbers of 2000 activated FimDCH, 487805 FimCA and, for reactions involving FimI, 244, 488, 976, 1463, 1951, 2439, 3415 or 8049 FimCI molecules. Using a concentration of 0.082 μM for activated FimDCH (the value obtained from the global fit of the experimental kinetic data) this corresponded to 20 μM FimCA, 0.01, 0.02, 0.04, 0.06, 0.08, 0.1, 0.14 or 0.33 μM FimCI, and a reaction volume of $4.05 \cdot 10^{-14}$ l. The rate constants used were $k_{on} = 1.1 \cdot 10^{7}$ M$^{-1}$ s$^{-1}$ and $k_{off} = 267$ s$^{-1}$ for binding/dissociation of FimCA to/from activated FimDCH; $k_{on} = 1.45 \cdot 10^{8}$ M$^{-1}$ s$^{-1}$ and $k_{off} = 94$ s$^{-1}$ for binding/dissociation of FimCI to/from activated FimDCH; $k_{(A-F)} = 5.5 \cdot 10^{-3}$ s$^{-1}$ for incorporation of the first FimA monomer when FimF is at the growing end of the pilus; $k_{(A-A)} = 1.4$ s$^{-1}$ for incorporation of FimA monomers when FimA is at the growing end of the pilus; $k_{(I-A)} = 0.14$ s$^{-1}$ for incorporation of FimI; and $k_{(A-A)}$ uncat $= 0.73$ M$^{-1}$ s$^{-1}$ for uncatalyzed polymerization of FimCA. Second-order rate constants were converted to their stochastic values by dividing by $N_A \cdot V$ (for binding of FimCA or FimCI to activated FimDCH) or by multiplying by $2/(N_A \cdot V)$ (for the uncatalyzed polymerization of FimCA), where $N_A$ is Avogadro's constant and V the reaction volume[79]. All rate constants were assumed to be independent of polymer length. The simulations were run for 65 min using the stochastic simulation algorithm[80] as solver and LogDecimation set to 10000. Pilus lengths were calculated by multiplying the number of FimA monomers assembled by each FimD molecule by 0.8 nm, the axial rise of the type 1 pilus rod[12]. Pili shorter than 15 nm were removed from the datasets and thus excluded from the comparison with the experimental length distributions as it was not possible to distinguish such short pili (if present) from background in the EM micrographs.

## Crystallization, data collection and structure determination

Crystals of the FimC$_{His}$-FimIt complex were obtained at 4 °C via sitting drop vapor diffusion by mixing 1.5 μl of protein solution (22 mg/ml in 10 mM Tris–HCl pH 8.0) with 1 μl of precipitant solution containing 4.5 M sodium formate, 0.1 M sodium cacodylate at pH 6.5. First crystals appeared after three days. Crystals were cryo-protected by adding 8 μl of 20% ethylene glycol in mother liquor solution directly to the drop and then flash-cooled in liquid nitrogen.

Crystals of the FimD$_N$-FimC$_{His}$-FimIt complex were obtained at 4 °C via sitting drop vapor diffusion by mixing 1 μl of protein solution (19.5 mg/ml in 10 mM Tris–HCl pH 8.0) with 1 μl of precipitant solution containing 0.1 M Hepes pH 8.4, 15 % PEG 4000. Crystals were cryo-protected in mother liquor containing 30% (v/v) ethylene glycol and flash-cooled in liquid nitrogen.

Diffraction data for FimC$_{His}$-FimIt and FimD$_N$-FimC$_{His}$-FimIt were collected at 100 K using a wavelength of 1 Å at beamline X06DA (Paul Scherrer Institute, Villigen, Switzerland) on a Pilatus 2MF pixel detector. For FimC$_{His}$-FimIt, data were processed to a final resolution of 1.75 Å with XDS[81] in space group P 31 2 1 with one complex per asymmetric unit. For FimD$_N$-FimC$_{His}$-FimIt, data were processed with XDS to a final resolution of 1.70 Å in space group P 21 21 21. The structures were solved by molecular replacement using PHASER[82]. For FimC$_{His}$-FimIt, the structure of the binary FimC-FimAt complex (pdb code 4DWH)[14] was used as a search model. For FimD$_N$-FimC$_{His}$-FimIt, the structure of the FimC$_{His}$-FimIt complex was used as a search model. After molecular replacement, initial refinement revealed the presence of FimD$_N$, which was then copied from PDB 1ze3. Refinements were performed with PHENIX[83], and COOT[84] was used for manual rebuilding of the model. Due to missing electron density, the following residues were omitted from the final models: for FimC$_{His}$-FimIt: FimI residues 123–127; for FimD$_N$-FimC$_{His}$-FimIt: FimD$_N$ residues 11–14, 124–125 and FimI residues 119–121.

Crystals of the FimC$_{His}$-FimI-FimAt$_{His}$ complex were obtained at 293 K via the vapor diffusion sitting drop method by mixing 2 µl of protein solution (10.2 mg/ml, containing less than 20 mM NaCl) with 2 µl of precipitant solution containing 0.8 M sodium formate, 100 mM Tris/acetic acid pH 8.5 and 16% PEG 4000 (w/v). Crystals grew to full size within 8 days. Diffraction data to 2.8 Å resolution were collected at 100 K on a Mar225 detector at beamline X06SA of the Swiss Light Source (PSI Villigen, Switzerland). For cryoprotection, a solution containing 16% PEG4000, 0.8 M sodium formate, 100 mM Tris-acetic acid pH 8.5, and 22% (v/v) ethylene glycol was used. Crystals were bathed in the cryo solution for 30–60 s before direct flash cooling in the cryostream. Data were processed with XDS[81] in space group P6(1). The structure was solved by molecular replacement with PHASER[82], using the structure of the FimC-FimAt complex (PDB file 3SQB)[14] as search model for FimC$_{His}$-FimI, and FimAt from the same complex as search model for the FimAt subunit in FimC$_{His}$-FimI-FimAt$_{His}$. Based on Matthews volume considerations the asymmetric unit was expected to contain at least two copies of the ternary complex. The PHASER runs could locate a complete copy of FimC$_{His}$-FimI-FimAt$_{His}$ and an additional FimC$_{His}$-FimI complex. The lacking FimAt molecule in the second copy of the ternary complex was placed manually by superimposing the first FimC$_{His}$-FimI-FimAt$_{His}$ complex onto the second FimC$_{His}$-FimI. The second FimAt molecule was found to fit into the electron density calculated from the molecules already located by PHASER. No further complexes could be located and we concluded that there are two FimC$_{His}$-FimI-FimAt$_{His}$ complexes in the asymmetric unit, corresponding to a Matthew coefficient of 4.64 A$^3$/Da (solvent content of 73.5%). This quite high solvent content is readily explained by the presence of large channels in the crystal packing. The structure was refined by successive rounds of manual model building with COOT[84] and refinement with PHENIX[83] using TLS groups defined by PHENIX and scale factors for the X-ray/stereochemistry weight (wxc_scale) and X-ray/ADP weight (wxu_scale) fixed to 0.25 and 1.66, respectively. Due to missing electron density, the following residues were omitted from the final model: residues 1–10 and 124–126 (chain B, FimI), residues 14–24, 57–69, 121–129 and 159 (chain C, FimAt), residues 1–5 and 123–127 (chain E, FimI) and residues 14–17 (chain F, FimAt).

Validation with MolProbity[85] showed that all final models possessed good stereochemical quality. Data collection and refinement statistics are shown in Supplementary Table 2. Structural superpositions of proteins and Cα RMSD values were calculated using SUPERPOSE and secondary-structure matching[86]. The interface area between FimI and FimAt in the FimC$_{His}$IA complex was calculated using PISA[87], all structural representations were created with PyMOL[88]. The structure-based sequence alignment of all type 1 pilus subunits was created with Expresso[89], using both SAP and TMalign for structural

alignment, and rendered using ESPript 3.0[90]. Secondary structure of FimI was assigned using DSSP[91,92].

### Reporting summary

Further information on research design is available in the Nature Portfolio Reporting Summary linked to this article.

### Data availability

Atomic coordinates were deposited in the Protein Data Bank (PDB) with accession codes 6SWH (FimC$_{His}$IA), 7B0W (FimC$_{His}$It) and 7B0X (FimD$_N$CI). Previously solved structures used in this study have PDB accession codes 4DWH, 1ZE3 and 3SQB. Source data are provided with this paper.

### Code availability

The complete MATLAB script used for Monte Carlo simulations of type 1 pilus rod assembly is provided as a Supplementary Software file.

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

## Acknowledgements

We thank the lab of Donald Hilvert and Peter Kast for generously providing plasmid pKTCTET-0; Gabriel Waksman for kindly providing plasmids pAN2-Strep and pETS1001; Marcel Bolten, Marc Leibundgut and Lena Keller for discussions; the Scientific Center for Optical and Electron Microscopy (ScopeM) of ETH Zurich for providing access to the electron microscope, and Peter Tittmann for technical support. We are grateful to Stephan Handschin from ScopeM for help with Fiji and TrakEM2; to Andrea Prota for help with refinement; to Arthur Goldsipe for help with stochastic simulations; to Jessica Stanisich for critical reading of the manuscript, and to Hiang Dreher-Teo and Helene Fäh-Rechsteiner for excellent technical assistance. This work was supported by grants 31003A_156304 and 310030B_176403/1 from the Swiss National Science Foundation to R.G.

## Author contributions

C.G. reconstituted pilus assembly, measured and evaluated kinetic data, performed electron microscopy, generated experimental and simulated pilus length distributions, performed in vivo titration of FimA and agglutination experiments, and designed experiments; C.P. performed and evaluated kinetic experiments; O.I. measured and evaluated binding of FimCI to $FimD_N$, determined $FimC_{His}$ dissociation rates and crystallized $FimC_{His}IA$; Z.B. investigated oxidative folding of FimI and FimA, performed real-time PCR experiments and crystallized $FimC_{His}It$ and $FimD_{N}CI$; M.E.W. measured kinetic data; M.A.S. and G.C. solved the structures of $FimC_{His}It$, $FimD_{N}CI$ and $FimC_{His}IA$; R.G. designed experiments and supervised the work; C.G. and R.G. wrote the paper, with contributions from C.P., O.I., Z.B. and M.A.S.

## Competing interests

The authors declare no competing interests.
