## [Peer Review File · Nature Communications]

Stochastic chain termination in bacterial pilus assemblyReviewer #1 (Remarks to the Author):

Giese et al. report the mechanism of termination of type 1 pilus assembly and pilus anchoring in the outer membrane, which they determined using the global kinetics analysis of pilus assembly and its termination by the FimI subunit, high resolution structural study of protein complexes corresponding to the FimI-incorporated pilus base, functional studies, and co-expression experiments. This is an interesting study that is overall well executed and well presented. Its findings will be of broad interest to the NSMB audience. At this time, there remain some points of concern that merit to be addressed in a revised manuscript:

Main points:

1. The authors' kinetic model for pilus assembly assumes that DSE at the usher is an irreversible process. Indeed, pili assembled via the chaperone-usher pathway (CUP) have extraordinary high stability (especially kinetic stability for unfolding-dissociation) as it was shown previously in the same lab (Puorger et al. *Structure* 2008 DOI: 10.1016/j.str.2008.01.013) and by Zavialov et al. (BJ 2005 DOI: 10.1042/BJ20050426) using in vitro assays. However, zip-out-zip-in DSE at the usher is sterically catalyzed and hence may have significantly lower kinetic barriers for both the forward and reverse reactions. Furthermore, the chaperone-subunit binding is sufficiently tight to compete with subunit-subunit binding. For example, the Caf1M-Caf1 chaperone-subunit pre-assembly complex of the F1 capsule of *Y. pestis* is characterized with $k_{on}=5.48 \pm 0.16 \times 10^5 \text{ M}^{-1} \text{ s}^{-1}$, $k_{off}=3.63 \pm 0.06 \text{ ms}^{-1}$, and $K_d=66.2 \pm 3.0 \text{ nM}$ (Yu et al. *JMB* 2012 DOI: 10.1016/j.jmb.2012.01.020). These facts argue that DSE should be considered as a reversible process in the model. Nevertheless, the model has reasonably good predictive power. I wonder if some other factors could have an effect. For example, the helical structure of the pilus rod on the other side of the usher may greatly reduce reversibility of the assembly at the usher. The role of the rod quaternary structure in pilus secretion has recently been highlighted (Pakharukova et al. *Nature* 2022 DOI: 10.1038/s41586-022-05095-0). Can the authors modify their model or explain why their assumption is valid?

2. The authors show that FimI-incorporated pilus has high affinity to the pilus-capping chaperone and suggest that this effect may determine the FimI-mediated pilus anchoring in the outer membrane. I fully agree with the proposed mechanism, but, in my opinion, this study requires an additional experiment. Zavialov et al. (BJ 2005) showed that, in the tertiary complex Caf1M-Caf1-Caf1, interactions between the Caf1M chaperone and the Caf1-Caf1 minimal fiber are much stronger than interactions between Caf1M and Caf1 in the Caf1M-Caf1 pre-assembly complex, and they proposed that the tighter fiber capping might be important for anchoring fibers in the outer membrane. Moreover, the melting temperature of this contact (55°C) coincides with the temperature typically used for the production surface-sheared pili by the heat treatment. It seems that in the Fim system, the FimA rod is bound much weaker to the chaperone such that the proper attachment of the pilus requires the specialized FimI subunit with higher affinity to the chaperone. Can the authors demonstrate this experimentally by comparing the k_{off} rates for the FimCAA and FimCIA complexes using their established assay? In my opinion, this comparison, is more relevant than comparison of k_{off} rates for FimCIA and FimCFG. While the structure of FimCAA is not known, structures of PapAA (Verger et al. 2007) and FimCA (author's lab) are available. I wonder if modeling based on these two structures and its comparison with FimCIA would be helpful to elucidate structural differences enabling tighter binding of FimC in the FimCIA complex?

Minor points:

1. In introduction. The statement "the periplasmic chaperone FimC that accelerates pilus subunit folding in the periplasm up to 104-fold" should be clarified. Comparison of the chaperone-bound subunit with its final pilus-inserted conformation shows that it not folded, but rather has a conformation of a molten globule-like folding intermediate, in which the hydrophobic core is not condensed (Zavialov et al. 2003 *Cell*, Zavialov et al. BJ 2005). This is especially clear in the case of chaperone-subunit pre-assembly complexes from the archaic and alternative CUPs. Chaperone-bound subunits in these systems have substantially unfolded conformations (Pakharukova et al. *PloS pathogens* 2015 DOI: 10.1371/journal.ppat.1005269, Pakharukova et al. *JBC* 2018 DOI:

10.1074/jbc.RA118.004170). Although periplasmic chaperones do assist in subunit folding as it has been demonstrated in the authors' lab (Vetsch et al. Nature 2004), it is important to point out that subunit folding on the chaperone is not complete. Besides, the completion of subunit folding drives fiber formation, which also worth mentioning.

2. In introduction. "DsbA catalyzes formation of an invariant structural disulfide bond connecting β -strands A and B in each of the subunits". The disulfide bond connecting β -strands A and B is highly conserved, but is not invariant. For example, it is missing in Caf1 (F1 capsule), MyfA (Myf), PsaA (pH6 antigen), CsaA and CsaB (CS6 colonization factor) etc. Interestingly, DSC-stabilized subunits from these systems are as stable as other CUP pilus subunits, which suggests that this disulfide bond is not essential for the exceptional stability of CUP pili.

3. In introduction. "Structural data suggest that the DSE reaction is initiated by threading the donor strand of the incoming subunit into the P5 pocket of the acceptor subunit, which is not occupied when the acceptor subunit is capped with FimC". It is worth mentioning that the P5 pocket is not always unoccupied. For example, in the Caf system, it is occupied by a Val residue of the G1 strand of the chaperone and is replaced by a Leu residue in Gd strand of the subunit donor strand in DSE (Zavialov et al. BJ 2005, Yu et al. JMB 2012). Moreover, in the archaic CUP Csu system, the DSE initiation site in the chaperone-bound subunit is not pre-folded (Pakharukova et al. PloS pathogens 2015, Pakharukova et al. JBC 2018). Since the authors report variations in the structure of the P5 in the Fim system (the P5 site is apparently not fully formed in FimA and is missing in FimI), a short discussion of differences in the structure of the P5 site in other systems, either in Introduction or Results and Discussion, would be helpful.

4. Page 6. "In contrast, formation of the chaperone-usher-adhesin complex alone appears to be sufficient for the assembly of pilus rods in pilus systems only bearing a single adhesin at the distal end of the pilus rod, such as pili from the Caf1, Afa/Dr or the CS1 family." This sentence contains incorrect information. In the Caf (F1 antigen) system, there is only one single-domain subunit Caf1 that forms the entire pilus. Similarly, Myf and Psa pili are homopolymers, in which polymerizing subunits act as adhesins (e.g. Pakharukova et al. Mol. Microbiol. doi.org/10.1111/mmi.13481). Afa/Dr belong to the same family of polyadhesins (also known as FGL or gamma-3 CUP family; Zavialov et al. FEMS Microbiol Rev 2007, Zav'yalov et al. Microbiol Rev 2010), but these pili, in addition, contain a single-domain tip subunit. It was suggested that, in gamma-3 CUP systems, pilus assembly is initiated by the tertiary complex (e.g. Caf1M-Caf1-Caf1) that is formed spontaneously in the periplasm (Zavialov et al. Cell 2003).

Reviewer #2 (Remarks to the Author):

The manuscript by Giese and coworkers reports the mechanism of chain termination of bacterial type 1 pili via FimI. The structure and mechanism of subunit incorporation in type 1 pili via FimD have been elucidated in the past but how this reaction is terminated and whether this is of functional relevance was unclear to date. Studies have suggested how pilus termination is achieved in the related P pili via PapH but a complete picture of assembly termination was lacking. The authors here present a comprehensive kinetic model of pilus assembly and assembly termination. They provide a detailed biochemical reconstitution of pilus assembly in vitro and show that FimI stabilizes the pilus in vivo. In addition, crystal structures of FimI in complex with FimC, FimDC and FimC and A are presented to get structural insights into how FimI terminates pilus assembly.

The manuscript is well-written, and all data are clearly presented. Especially the part on how the kinetic model is established and used for simulations is meticulously written with strong attention given to the details. This is reflected by the abstract and title, which clearly focus on the quantitative model but do not include that chain termination is actually very relevant in vivo and contributes to pilus anchoring and stability. I suggest that the authors mention their functional and structural data on FimI in the abstract already. Overall, this study provides comprehensive insights into the role of FimI and thereby closes a gap in the understanding of pili assembly. I have

outlined some more issues below that should be addressed prior to publication.

Page 2, Line 5: The abstract should not contain references.

Page 15, line 28: Figure 4C is mentioned earlier than 4A and B in the text. Please change.

Page 15, Line 34: Could you elaborate why the CDS was not removed entirely? Does this change the operon architecture and prevent transcription of downstream genes? There are obviously slight changes in the gene expression of fimC as shown in figure S5C.

Page 18, Line 17: Can the authors comment on how pilus length correlates with its stability? Is there an optimal pilus length regarding the yeast agglutination assays shown in figure 5? Does the fimA strain complemented with pCG1-AC behave like the WT in the presence of 7.5 ng/ml aTc?

Page 18, line 30: I do not see a full rescue of the fimI deletion phenotype here. Rather a partial rescue. Why did the authors use a comparatively high aTc concentration of 30 ng/ml, while they only used a maximum of 17.5 ng/ml for fimA induction? Is it possible that the partial rescue is a result of an imbalance between FimI and FimD leading to shorter pili that are less efficient in yeast binding? A real-time PCR might be helpful here to judge this.

Page 20, Line 3: Which cell densities were used for the experiments in figure 5A?

Page 20, Line 33: "...were incubated in the presence..."

Page 23, Line 19-25: The authors should provide a figure here. It's difficult to follow this paragraph.

Figure 7F: It's difficult to see whether the P5 pocket is open or closed. The authors might want to show the pocket in a closeup with the nearby elements labeled.

Figure S3C: Please label the lines with the respective concentrations of FimDN instead of only referring to the figure legend.

ETH Zurich
Dr. Christoph Giese
HPK E2
Otto-Stern-Weg 5
8093 Zurich, Switzerland

Phone +41 44 633 32 92
giesec@mol.biol.ethz.ch

Zurich, 24 August 2023

Please find our detailed point-by-point answers to the reviewer comments below.

Reviewer 1:

Giese et al. report the mechanism of termination of type 1 pilus assembly and pilus anchoring in the outer membrane, which they determined using the global kinetics analysis of pilus assembly and its termination by the FimI subunit, high resolution structural study of protein complexes corresponding to the FimI-incorporated pilus base, functional studies, and co-expression experiments. This is an interesting study that is overall well executed and well presented. Its findings will be of broad interest to the NSMB audience. At this time, there remain some points of concern that merit to be addressed in a revised manuscript:

We would like to thank the reviewer for the positive comment and appreciate the very constructive and comprehensive review of our work.

Main points:

1. The authors' kinetic model for pilus assembly assumes that DSE at the usher is an irreversible process. Indeed, pili assembled via the chaperone-usher pathway (CUP) have extraordinary high stability (especially kinetic stability for unfolding-dissociation) as it was shown previously in the same lab (Puorger et al. Structure 2008 DOI: 10.1016/j.str.2008.01.013) and by Zavialov et al. (BJ 2005 DOI: 10.1042/BJ20050426) using in vitro assays. However, zip-out-zip-in DSE at the usher is sterically catalyzed and hence may have significantly lower kinetic barriers for both the forward and reverse reactions. Furthermore, the chaperone-subunit binding is sufficiently tight to compete with subunit-subunit binding. For example, the Caf1M-Caf1 chaperone-subunit pre-assembly complex of the F1 capsule of *Y. pestis* is characterized with $k_{on}=5.48\pm 0.16 \times 10^5 \text{ M}^{-1} \text{ s}^{-1}$, $k_{off}=3.63\pm 0.06 \text{ ms}^{-1}$, and $K_d=66.2\pm 3.0 \text{ nM}$ (Yu et al. JMB 2012 DOI: 10.1016/j.jmb.2012.01.020). These facts argue that DSE should be considered as a reversible process in the model. Nevertheless, the model has reasonably good predictive power. I wonder if some other factors could have an effect. For example, the helical structure of the pilus rod on the other side of the usher may

greatly reduce reversibility of the assembly at the usher. The role of the rod quaternary structure in pilus secretion has recently been highlighted (Pakharukova et al. Nature 2022 DOI: 10.1038/s41586-022-05095-0). Can the authors modify their model or explain why their assumption is valid?

We thank the reviewer for raising this question. We agree that donor strand exchange is sterically catalyzed by FimD. However, the barrier for pilus disassembly is extraordinarily high: The extrapolated half-life for pilus rod dissociation/unfolding at pH 2.0 is $4.8 \cdot 10^{21}$ years (Puorger et al. Structure 2008 DOI: 10.1016/j.str.2008.01.013). Donor strand exchange can therefore be considered irreversible. In addition, we note that reversibility of DSE has not been observed experimentally. We have added an additional sentence to our manuscript that addresses the role of quaternary structure formation of the assembled pilus rod in facilitating irreversibility of DSE and cite the work of Pakharukova et al. 2022 (see page 9, lines 23-25).

With respect to the kinetic model, we note that adjusting it to account for reversibility of donor strand exchange would require including three additional rate constants: for dissociation of FimCA from incorporated FimF, FimCA from incorporated FimA and FimCI from incorporated FimA. However, with these three additional parameters the fit becomes underdetermined and no longer provides reliable values for the fitted rate constants.

2. The authors show that FimI-incorporated pilus has high affinity to the pilus-capping chaperone and suggest that this affect may determine the FimI-mediated pilus anchoring in the outer membrane. I fully agree with the proposed mechanism, but, in my opinion, this study requires an additional experiment. Zavialov et al. (BJ 2005) showed that, in the tertiary complex Caf1M-Caf1-Caf1, interactions between the Caf1M chaperone and the Caf1-Caf1 minimal fiber are much stronger than interactions between Caf1M and Caf1 in the Caf1M-Caf1 pre-assembly complex, and they proposed that the tighter fiber capping might be important for anchoring fibers in the outer membrane. Moreover, the melting temperature of this contact (55oC) coincides with the temperature typically used for the production surface-sheared pili by the heat treatment. It seems that in the Fim system, the the FimA rod is bound much weaker to the chaperone such that the proper attachment of the pilus requires the specialized

FimI subunit with higher affinity to the chaperone. Can the authors demonstrate this experimentally by comparing the k-off rates for the FimCAA and FimCIA complexes using their established assay? In my opinion, this comparison, is more relevant than comparison of k-off rates for FimCIA and FimCFG. While the structure of FimCAA is not known, structures of PapAA (Verger et al. 2007) and FimCA (author's lab) are available. I wonder if modeling based on these two structures and its comparison with FimCIA would be helpful to elucidate structural differences enabling tighter binding of FimC in the FimCIA complex?

We thank the reviewer for the valuable comment and the suggestion to determine the FimC dissociation rate constant also for the FimCAA complex. We had not done this experiment initially because we assumed that a homogeneous FimCAA preparation would be very difficult to obtain. However, we now succeeded in establishing a reliable protocol for the production of FimCAA by incubating the FimC-FimA complex with a 5-fold excess of the FimC-FimA complex at 37 °C overnight and subsequent FimCAA purification by cation exchange and size exclusion chromatography. The FimC dissociation rate constant was then determined for the ternary FimCAA complex and, as a direct reference, the binary FimCA complex with our established competition experiment by incubating 4 μM FimCAA or FimCA with 36 μM FimC_{His} and recording the kinetics of relaxation to the new equilibrium.

Similar to our results obtained for FimI and in agreement with the data for Caf1 mentioned by the reviewer, FimC dissociated 40 times slower from FimCAA than from FimCA (dissociation half-lives were 18 min and 26 s, respectively). Nevertheless, the dissociation of FimC from FimCIA still was 13 times slower

(dissociation half-life of 3.9 h) than the dissociation of FimC from FimCAA, fully consistent with FimC's function as pilus assembly inhibitor and anchor in the outer membrane.

We summarized these results in a new supplementary figure in our manuscript (Figure S8) and updated main text and experimental section with respect to the production of FimCAA and the determination of its FimC dissociation rate accordingly (see page 22, lines 10-21; page 30, lines 15-20; page 33, lines 33-34; page 34 lines 1-9; page 36, lines 18-30; page 37 lines 13, 15, 26-34; pages 54, 55). In addition, we cite the work of Zavialov and co-workers (BJ 2005 DOI: 10.1042/BJ20050426) in our manuscript.

We used AlphaFold2 to predict the structure of FimCAA and analyzed this model using PISA. In comparison to the crystal structure of FimCIA, an analogous, 402 Å² FimC-FimAt interface is detected in the FimCAA model. Hydrophobic contacts between FimC and FimA are similar to those in FimCIA, for example between Ala195 of FimC and Pro145 of FimA. In addition, both proteins interact via amide- π stacking between the Gly146-Ala147 peptide bond of FimA and Tyr193 of FimC. As this contact is also present in the FimC-FimA interface of FimCIA but had not been pointed out specifically in the manuscript, we updated main text and Supplementary Figure S9E accordingly (see page 25, lines 11 & 12 and page 56). In contrast to FimCIA, the predicted FimC-FimA interface of FimCAA, however, lacked intermolecular hydrogen bonds and salt bridges. Together, these structural features of FimCAA may explain why FimC dissociates slower from FimCAA than FimCA, but not as slow as from FimCIA. We provide a figure of the predicted FimCAA structure below but prefer to not include it in the manuscript because of its theoretical nature.

Minor points:

1. In introduction. The statement “the periplasmic chaperone FimC that accelerates pilus subunit folding in the periplasm up to 10⁴-fold” should be clarified. Comparison of the chaperone-bound subunit with its final pilus-inserted conformation shows that it not folded, but rather has a conformation of a molten globule-like folding intermediate, in which the hydrophobic core is not condensed (Zavialov et al. 2003 Cell, Zavialov et al. BJ 2005). This is especially clear in the case of chaperone-subunit pre-assembly complexes from the archaic and alternative CUPs. Chaperone-bound subunits in these systems have substantially unfolded conformations (Pakharukova et al. PloS pathogens 2015 DOI: 10.1371/journal.ppat.1005269, Pakharukova et al. JBC 2018 DOI: 10.1074/jbc.RA118.004170). Although periplasmic chaperones do assist in subunit folding as it has been demonstrated in the authors’ lab (Vetsch et al. Nature 2004), it is important to point out that subunit folding on the chaperone is not complete. Besides, the completion of subunit folding drives fiber formation, which also worth mentioning.

We agree with the reviewer in that the chaperone traps the subunit in an intermediate, partially unfolded or near-native folding state in archaic and alternative chaperone-usher pilus systems. This is however not the case for FimC-subunit complexes from type 1 pili: All solved FimC-subunit structures indicate that the bound subunit has an intact tertiary structure (except for the N-terminal donor strand) (see for example Choudhury et al. Science 1999, Nishiyama et al. EMBO J. 2005 doi:10.1038/sj.emboj.7600693, Eidam et al. FEBS Letters 2008 doi:10.1016/j.febslet.2008.01.030, Crespo et al, Nat. Chem. Biol. 2012 doi: 10.1038/nchembio.1019, and the structures of FimCI and FimD_NCI in our current manuscript). Thus, in the case of type 1 pili, chaperone-bound subunits have a native tertiary structure, and adopt a more stable, alternative native tertiary structure in the context of the quaternary structure of the pilus. In addition, the at least 10⁴-fold acceleration of FimA folding by FimC was recorded by detection of native FimA molecules (Crespo et al, Nat. Chem. Biol. 2012 doi: 10.1038/nchembio.1019).

To address the point raised by the reviewer, we now specify that the up to 10⁴-fold acceleration of subunit folding by FimC applies to the type 1 pilus system and additionally mention that chaperone-bound subunits may not be fully folded in the other pilus systems mentioned by the reviewer (see page 3, lines 10-17).

2. In introduction. “DsbA catalyzes formation of an invariant structural disulfide bond connecting β -strands A and B in each of the subunits”. The disulfide bond connecting β -strands A and B is highly conserved, but is not invariant. For example, it is missing in Caf1 (F1 capsule), MyfA (Myf), PsaA (pH6 antigen), CssA and CssB (CS6 colonization factor) etc. Interestingly, DSC-stabilized subunits from these systems are as stable as other CUP pilus subunits, which suggests that this disulfide bond is not essential for the exceptional stability of CUP pili.

Our statement only referred to the type 1 pilus system, but we thank the reviewer for the clarification. We now removed “invariant” and changed the respective phrase to “...DsbA catalyzes formation of a structural disulfide bond connecting β -strands A and B in each of the type 1 pilus subunits...” (see page 3, lines 23,24).

3. In introduction. “Structural data suggest that the DSE reaction is initiated by threading the donor strand of the incoming subunit into the P5 pocket of the acceptor subunit, which is not occupied when the acceptor subunit is capped with FimC”. It is worth mentioning that the P5 pocket is not always unoccupied. For example, in the Caf system, it is occupied by a Val residue of the G1 strand of the chaperone and is replaced by a Leu residue in Gd strand of the subunit donor strand in DSE (Zavialov et al. BJ 2005, Yu et al. JMB 2012). Moreover, in the archaic CUP Csu system, the DSE initiation site in the chaperone-bound subunit is not pre-folded (Pakharukova et al. PloS pathogens 2015, Pakharukova et al. JBC 2018). Since the

authors report variations in the structure of the P5 in the Fim system (the P5 site is apparently not fully formed in FimA and is missing in FimI), a short discussion of differences in the structure of the P5 site in other systems, either in Introduction or Results and Discussion, would be helpful.

We have included additional information on whether or not the P5 pocket of an acceptor subunit is occupied by the donor strand of the chaperone in the introduction, and updated corresponding citations accordingly (page 4, lines 9-15).

4. Page 6. "In contrast, formation of the chaperone-usher-adhesin complex alone appears to be sufficient for the assembly of pilus rods in pilus systems only bearing a single adhesin at the distal end of the pilus rod, such as pili from the Caf1, Afa/Dr or the CS1 family." This sentence contains incorrect information. In the Caf (F1 antigen) system, there is only one single-domain subunit Caf1 that forms the entire pilus. Similarly, Myf and Psa pili are homopolymers, in which polymerizing subunits act as adhesins (e.g. Pakharukova et al. Mol. Microbiol. doi.org/10.1111/mmi.13481). Afa/Dr belong to the same family of polyadhesins (also known as FGL or gamma-3 CUP family; Zavialov et al. FEMS Microbiol Rev 2007, Zav'yalov et al. Microbiol Rev 2010), but these pili, in addition, contain a single-domain tip subunit. It was suggested that, in gamma-3 CUP systems, pilus assembly is initiated by the tertiary complex (e.g. Caf1M-Caf1-Caf1) that is formed spontaneously in the periplasm (Zavialov et al. Cell 2003).

We thank the reviewer for this important correction. We rephrased the respective sentence, added one additional sentence on the initiation of gamma-3 chaperone-usher pilus assembly and updated citations accordingly (see page 7, lines 28-32).

Reviewer #2:

The manuscript by Giese and coworkers reports the mechanism of chain termination of bacterial type 1 pili via FimI. The structure and mechanism of subunit incorporation in type 1 pili via FimD have been elucidated in the past but how this reaction is terminated and whether this is of functional relevance was unclear to date. Studies have suggested how pilus termination is achieved in the related P pili via PapH but a complete picture of assembly termination was lacking. The authors here present a comprehensive kinetic model of pilus assembly and assembly termination. They provide a detailed biochemical reconstitution of pilus assembly in vitro and show that FimI stabilizes the pilus in vivo. In addition, crystal structures of FimI in complex with FimC, FimDC and FimC and A are presented to get structural insights into how FimI terminates pilus assembly.

The manuscript is well-written, and all data are clearly presented. Especially the part on how the kinetic model is established and used for simulations is meticulously written with strong attention given to the details. This is reflected by the abstract and title, which clearly focus on the quantitative model but do not include that chain termination is actually very relevant in vivo and contributes to pilus anchoring and stability. I suggest that the authors mention their functional and structural data on FimI in the abstract already. Overall, this study provides comprehensive insights into the role of FimI and thereby closes a gap in the understanding of pili assembly. I have outlined some more issues below that should be addressed prior to publication.

We thank the reviewer for the positive feedback and now mention the functional and structural data on FimI in the abstract (see page 2, lines 13-17).

Page 2, Line 5: The abstract should not contain references.

References were removed from the abstract.

Page 15, line 28: Figure 4C is mentioned earlier than 4A and B in the text. Please change.

We now refer to the panels of Figure 4 as suggested by the reviewer and updated the text, Figure 4 and its legend accordingly (see page 16, line 30; page 17, line 4; page 18; page 19, lines 2,3 and 5).

Page 15, Line 34: Could you elaborate why the CDS was not removed entirely? Does this change the operon architecture and prevent transcription of downstream genes? There are obviously slight changes in the gene expression of *fimC* as shown in figure S5C.

The design of the *fimA* deletion strain essentially followed the one described in Baba et al. *Molecular Systems Biology* 2006 doi:10.1038/msb4100050. In this study, a set of *E. coli* K12 strains with in-frame, single-gene deletions of all non-essential genes is reported (so-called “Keio collection”). The initiating methionine and last six C-terminal amino acids of the target gene are retained. Not deleting the last 18 nucleotides of the target gene had the purpose of preventing interference with translation initiation of the downstream gene. Hence, the operon architecture is not perturbed. With respect to the *fimC* transcript levels measured in W3110 and W3110 Δ *fimA* cells (Figure S5H), these are identical within experimental error.

Page 18, Line 17: Can the authors comment on how pilus length correlates with its stability? Is there an optimal pilus length regarding the yeast agglutination assays shown in figure 5? Does the *fimA* strain complemented with pCG1-AC behave like the WT in the presence of 7.5 ng/ml aTc?

Whether or not the stability of the pilus, for example its kinetic stability against dissociation/unfolding, increases with increasing pilus length is currently unknown. Unfolding of the pilus rod and dissociation into monomers were shown to be coupled processes and could only be achieved with high denaturant concentrations at acidic pH, independently of pilus length (Puorger et al. *Structure* 2008 DOI: 10.1016/j.str.2008.01.013). In addition, one may consider the pilus rod to be conceptually similar to a repeat protein. For ankyrin repeat proteins, an increase of their thermodynamic, thermal and kinetic stability with increasing repeat number was indeed demonstrated (for example in Wetzel et al. *JMB* 2008 doi:10.1016/j.jmb.2007.11.046). We therefore cannot exclude a similar mechanism for the type 1 pilus rod. However, it is important to note that experimentally verifying this prediction, for example by measuring rate constants for denaturant-induced unfolding/dissociation as described in Puorger 2008, is likely very challenging and may not be possible at all. This is because in this type of experiment, rate constants measured at very high denaturant concentration (the only condition where these parameters are in fact experimentally accessible) are extrapolated back to a denaturant concentration of zero. Given that the median pilus lengths of different pilus preparations in our current manuscript differ by less than one order of magnitude, this length difference is likely insufficient to detect differences in kinetic stability. At the same time, protocols for producing shortest possible pilus rods with, for example, two helical turns and consisting of 6-7 copies of FimA are currently not available.

Regarding the second question of the reviewer, we note that the purpose of the yeast agglutination experiment was not to identify conditions that would result in an optimal pilus length distribution for the assay but to address FimI’s function of anchoring the pilus to the outer membrane, i.e., compare cells that had experienced shear stress with those that had not. We fully agree with the reviewer that for the efficiency of agglutination itself, an optimal pilus length distribution should exist. In this context,

W3110Δ*fimA* cells harboring plasmid pCG1-AC and grown in presence of 7.5 ng/ml aTc may indeed show a WT-like agglutination phenotype.

Page 18, line 30: I do not see a full rescue of the *fimI* deletion phenotype here. Rather a partial rescue. Why did the authors use a comparatively high aTc concentration of 30 ng/ml, while they only used a maximum of 17.5 ng/ml for *fimA* induction? Is it possible that the partial rescue is a result of an imbalance between FimI and FimD leading to shorter pili that are less efficient in yeast binding? A real-time PCR might be helpful here to judge this.

We agree with the reviewer that plasmid-encoded *fimI* only partially rescued *fimI* deficiency of W3110Δ*fimI* cells. The relevant section now reads "...plasmid-encoded FimI partially complemented..." (see page 19, line 31).

The difference in aTc concentrations used for complementation of the *fimI* deletion on the one hand and tuning the expression level of FimCA on the other is caused by differences in copy number of the plasmids used in the two experiments. While pCG1-AC is a low-copy number plasmid with p15A origin of replication, pKTCTET-IC has much higher copy number (pUC origin of replication) and, therefore, requires higher aTc concentrations than pCG1-AC for titration.

With respect to possible explanations for only achieving partial complementation of the *fimI* deletion phenotype when using plasmid-encoded FimCI, we note that there may be very many different reasons for this observation, as, for example, the plasmid also encodes FimC in addition to FimI. A potential imbalance between FimI and FimD levels, as suggested by the reviewer, is certainly one possibility. However, as the main purpose of this control experiment was to demonstrate complementation, we considered the potential reasons for only achieving partial complementation not relevant for the overall experiment.

Page 20, Line 3: Which cell densities were used for the experiments in figure 5A?

For the experiments shown in Figure 5A, the *E. coli* cells were diluted to $OD_{600} = 1.0$ prior to mixing with the yeast cells. We have included this information in the legend of Figure 5A (see page 20, line 5-6).

Page 20, Line 33: "...were incubated in the presence..."

We have corrected the sentence as suggested (see page, 21, line 33).

Page 23, Line 19-25: The authors should provide a figure here. It's difficult to follow this paragraph.

We prepared a new supplementary figure that shows how FimC and FimI interact in the FimC_{HisI}A complex (Figure S9A, page 56) and updated the figure legend accordingly (page 57, lines 2-5).

Figure 7F: It's difficult to see whether the P5 pocket is open or closed. The authors might want to show the pocket in a closeup with the nearby elements labeled.

To identify the state of the P5 pocket (open vs. closed) in the structures of the different pilus assembly intermediates in an unambiguous and unbiased manner, we analyzed them by CASTp, as described in our manuscript. According to this analysis, the P5 pocket is found in an open conformation in FimH_p of the FimD_NCH_p complex and in FimF of the FimDHGFC complex, while no pocket was identified in FimI, PapH and FimA of all other structures analyzed (described in the legend of Figure 7F). Corresponding atoms of

FimH_p and FimF that define the pocket are highlighted in the current version of Figure 7F in white. In addition, we indicated the state of the P5 pocket for every analyzed structure in each panel of Figure 7F. Therefore, we think that the current version of Figure 7F is very clear about the P5 pocket conformation in the different complexes and do not see possibilities for improving it further.

Figure S3C: Please label the lines with the respective concentrations of FimDN instead of only referring to the figure legend.

We have now included the concentrations of FimCI in Figure S3C as suggested (see page 48).

We hope that we could adequately address the points raised by the reviewers.

Thank you very much again for considering our manuscript for publication.

Yours sincerely,

Christoph Giese

Reviewer #2 (Remarks to the Author):

I thank the authors for addressing all of my questions and concerns. The manuscript has significantly improved and I recommend publication in its current form.